# A non-cell-autonomous role for Pml in the maintenance of leukemia from the niche

Jlenia Guarnerio[1], Lourdes Maria Mendez[1], Noboru Asada[2], Archita Venugopal Menon[1], Jacqueline Fung[1], Kelsey Berry[1], Paul S. Frenette[2], Keisuke Ito [2] & Pier Paolo Pandolfi [1]

Disease recurrence after therapy, due to the persistence of resistant leukemic cells, represents a fundamental problem in the treatment of leukemia. Elucidating the mechanisms responsible for the maintenance of leukemic cells, before and after treatment, is therefore critical to identify curative modalities. It has become increasingly clear that cell-autonomous mechanisms are not solely responsible for leukemia maintenance. Here, we report a role for Pml in mesenchymal stem cells (MSCs) in supporting leukemic cells of both CML and AML. Mechanistically, we show that Pml regulates pro-inflammatory cytokines within MSCs, and that this function is critical in sustaining CML-KLS and AML ckit[+] leukemic cells non-cell autonomously.

[1] Cancer Research Institute, Beth Israel Deaconess Cancer Center, Departments of Medicine and Pathology, Beth Israel Deaconess Medical Center, Harvard Medical School, Boston, MA 02215, USA. [2] Ruth L. and David S. Gottesman Institute for Stem Cell and Regenerative Medicine Research, Department of Medicine and Cell Biology, Albert Einstein College of Medicine, Michael F. Price Center, 1301 Morris Park Avenue, Bronx, NY 10461, USA. Lourdes Maria Mendez and Noboru Asada contributed equally to this work. Correspondence and requests for materials should be addressed to P.P.P. (email: ppandolf@bidmc.harvard.edu)

Relapsed disease, following complete remission achieved under conventional or targeted therapy, remains as a central problem in the treatment of leukemia[1]. Because of their resistance to cytotoxic therapy, it was hypothesized that leukemia-initiating cells (LICs, also identified as leukemia stem cells LSCs) could be the cells that are responsible for the relapse of leukemia[2], highlighting the need for novel therapeutic strategies that specifically target this population of cells.

The phenotype of LICs has been characterized in chronic myeloid leukemia (CML) and several subtypes of acute myeloid leukemia (AMLs)[3–5], but the mechanisms responsible for the maintenance of LICs are not yet fully elucidated, and have largely focused on signals intrinsic to leukemic cells, i.e., cell-autonomous mechanisms. However, it has become apparent that, in addition to the cell-autonomous mechanisms, non-cell-autonomous factors are also critically responsible for the persistence of LICs[6–10]. Moreover recent data are showing that LICs are also able to manipulate the composition of the bone marrow (BM), triggering functional changes in normal HSCs, as well in mesenchymal stem cell (MSCs)[11–13]. The dynamic interplay between leukemic cells and stromal cells has been described in different types of myeloid leukemia, ranging from CML to AML[5,14]. These observations suggest that elucidating the mechanisms by which MSCs maintain LICs might reveal new therapeutic approaches that could be combined with current therapies in the attempt to definitively treat several types of myeloid leukemia.

The Promyelocytic leukemia (PML) gene was identified as a result of its involvement in the t(15;17) chromosomal translocation of acute promyelocytic leukemia (APL)[15–17], and has since been extensively studied in cancer mainly as a tumor suppressor gene, which controls fundamental biological processes such as apoptosis, cellular proliferation, and senescence[18,19]. However, and contrary to its established tumor suppressive role, we have also found that in CML, higher levels of Pml correlate with a poor clinical outcome and a decreased frequency of complete molecular response. Genetic and pharmacologic ablation of Pml, using arsenic trioxide ($AS_2O_3$), lead CML-LICs to exit from quiescence and early expansion, followed by their exhaustion, and thus to the eradication of the disease[20]. Additionally, we have described that the PML/PPAR/FAO pathway is essential in the maintenance of HSCs, by regulating their asymmetric cell division[21]. However, the roles of Pml in the BM microenvironment are not yet defined.

We here identify a novel non-cell-autonomous role and mechanism by which Pml expressed in MSCs sustains leukemic cells, with possible therapeutic implications for both chronic and acute myeloid leukemia.

## Results

**Pml regulates MSCs biology**. We have previously reported that the loss of Pml triggers HSCs and CML-LICs to exit from quiescence, expand, and eventually exhaust[20]. Interestingly, when we analyzed the BM composition of Pml-null mice, as compared to wild-type mice, we also noted that bone-associated MSCs (the compartment enriched in the population of PαS cells: CD45$^-$CD31$^-$Ter119$^-$CD51$^+$PDGFRα$^+$Sca1$^+$ [22,23]), expanded upon Pml loss, while there was no difference in the total number of stromal cells (defined as CD45$^-$CD31$^-$Ter119$^-$CD51$^-$PDGFRα$^-$Sca1$^-$) (Fig. 1a). In order to understand if the initial expansion of MSCs was followed by exhaustion, similar to what was previously observed for the Pml$^{-/-}$ HSCs[20], we analyzed the mesenchymal composition of the BM of older mice (15 months old). As shown in Fig. 1b, although the number of MSCs was reduced overall during aging, 15-month-old Pml$^{-/-}$ mice still showed more PαS cells in crushed BM tissues, as compared to 15-month-old Pml$^{+/+}$ control mice, suggesting that the early expansion of PαS cells upon loss of Pml, is not followed by terminal exhaustion.

Next, to investigate whether the expansion of MSCs was orchestrated by cell-autonomous mechanisms, we generated and analyzed conditional knockout mice, where the expression of Pml was selectively deleted in the mesenchymal compartment. To this end, we first generated Pml-floxed (Pml$^{F/F}$) mice (see "Methods" section and Supplementary Fig. 1a), and crossed them with Prrx1-Cre mice. In the Prrx1-Cre-Pml$^{F/F}$ mice, the expression of Pml was significantly reduced in MSCs (Supplementary Fig. 1b); accordingly, and similar to the total body Pml-null mice, Prrx1-Cre-Pml$^{F/F}$ mice showed significantly increased numbers of MSCs, when compared to the controls, while no differences were observed in the total number of CD45$^-$CD31$^-$Ter119$^-$CD51$^-$PDGFRα$^-$Sca1$^-$stromal cells (Fig. 1c). Taken together, these results suggest that Pml is functional in the mesenchymal compartment of the BM where it reduces the expansion of CD45$^-$CD31$^-$Ter119$^-$CD51$^+$PDGFRα$^+$Sca1$^+$ MSCs.

To investigate more thoroughly the role of Pml in MSCs, we performed in vitro experiments to assess the clonogenic ability of MSCs, their proliferation rate, and their differentiation potential. MSCs from Pml$^{+/+}$ or Pml$^{-/-}$ mice were seeded in culture (in selected experiments, cells were also isolated from the Prrx1-Cre-Pml$^{F/F}$ conditional knockout mice) and kept in hypoxic conditions as previously reported[23]. The cells' capacity to form CFU-F colonies was then measured after 5 days in culture. Regardless of Pml expression status, cells showed similar morphology and a comparable ability to form CFU-F colonies (Fig. 1d and Supplementary Fig. 1c). Although no differences in growth were detected while culturing the cells at early passages, Pml$^{-/-}$ cells showed a higher proliferation rate, compared to Pml$^{+/+}$ cells, if grown for several passages (Supplementary Fig. 1d). At any given time during the culture, no signs of spontaneous differentiation into adipocytes or osteoblasts were observed (Fig. 1e). Irrespective of Pml expression, MSCs showed similar potential for differentiation into cells of the mesenchymal lineage, once they were triggered to differentiate with specific factors (Fig. 1f and Supplementary Fig. 1e, f).

Collectively, in vitro and in vivo experiments show that Pml is functional in MSCs, but it does not confer CFU-F clonogenic potential to the cells, and it does not seem to have an effect on the capacity of MSCs to differentiate into the downstream progeny.

**Pml only marginally sustains HSCs non-cell autonomously**. Recent publications indicate that the maintenance of HSCs is coordinated by signals originating within, and also outside of, the cell. In this respect, MSCs have been described as extrinsic players in the maintenance of HSCs[24]. Since we observed a role for Pml in MSCs, we next investigated the possibility that Pml could regulate HSCs in a non-cell-autonomous manner through MSCs. To this end, we first analyzed the HSC compartment of Prrx1-Cre-Pml$^{F/F}$ mice. As shown in Fig. 2a and Supplementary Fig. 2a, the absence of Pml in MSCs did not affect the overall number of hematopoietic stem/progenitor cells, leaving the total number of SLAM$^+$CD48$^-$KLS (Lin$^-$ckit$^+$Sca1$^+$), CD34$^+$ KLS, and KLS cells in the BM of mice virtually unchanged.

In order to investigate a possible role of Pml in mediating interactions between MSCs and HSCs, we analyzed the three-dimensional distribution of HSCs in the BM of Prrx1-Cre-Pml$^{F/F}$ mice compared to controls, using an approach that combines whole-mount confocal immunofluorescence imaging techniques and computational models[25]. By performing this analysis, we found a slight but significant alteration of HSC distribution with a shift in distribution closer to arterioles in Prrx1-Cre-Pml$^{F/F}$ mice

compared to controls (Fig. 2b). However, when we next co-cultured in vitro MSCs with HSCs to directly assess the non-cell-autonomous capacity of Pml to sustain HSCs (Fig. 2c), we did not notice any substantial differences in terms of HSCs maintenance by comparing the co-cultures based on $Pml^{+/+}$ or $Pml^{-/-}$ MSCs. As shown in Fig. 2d, during the co-cultures, the HSCs had comparable times during which differentiation occurred, regardless of whether they were placed onto $Pml^{+/+}$ or $Pml^{-/-}$ MSCs. Collectively, the analysis of the BM composition of the Pml-conditional knockout mice, and the in vitro co-cultures suggest no major function of Pml expressed in MSCs for the HSC maintenance within the BM.

The maintenance of LSCs, like that of HSCs, is also coordinated by cell-autonomous and non-cell-autonomous signals[26]. We have previously reported that upon treatment with $AS_2O_3$, CML-LSCs were exhausted to a greater extent than normal HSCs[20]. These findings, along with the observation of the cell-autonomous role of Pml in MSCs, led us to hypothesize that Pml could play a non-cell-autonomous role in the regulation of CML-LSCs through MSCs. In order to investigate this possibility, we performed serial transplantation experiments using either $Pml^{-/-}$ or $Pml^{+/+}$ mice as recipients, as described in Supplementary Fig. 2b. Once first recipient mice developed more than 50% of leukemic blasts in the peripheral blood (both for $Pml^{-/-}$ or $Pml^{+/+}$ mice, data not shown), $BA^+GFP^+$ cells were collected from the BM and re-transplanted in equal numbers into the secondary recipient $Pml^{-/-}$ or $Pml^{+/+}$ mice. These mice were then analyzed for the presence of $BA^+GFP^+$ KLS ($ckit^+Lin^-Sca1^+$) cells. Interestingly, the number of $BA^+GFP^+$ KLS cells was reduced more than twofold in $Pml^{-/-}$ mice, as compared to $Pml^{+/+}$ mice, whether the $BA^+GFP^+$ KLS cells were harvested from the endosteal surface or from the inner cavity of the BM (Supplementary Fig. 2b). In order to further corroborate these results, we next performed serial transplantation experiments in Pml conditional mice. To this end, we transplanted $BA^+GFP^+$ leukemic cells in $Prrx1$-$Cre$-$Pml^{F/F}$ mice, or $Prrx1$-$Cre$-$Pml^{+/+}$ mice as control, and we then analyzed the survival rate of the recipients. No differences were detected in the survival rate of the two cohorts of recipients at first transplantation. However, secondary recipient

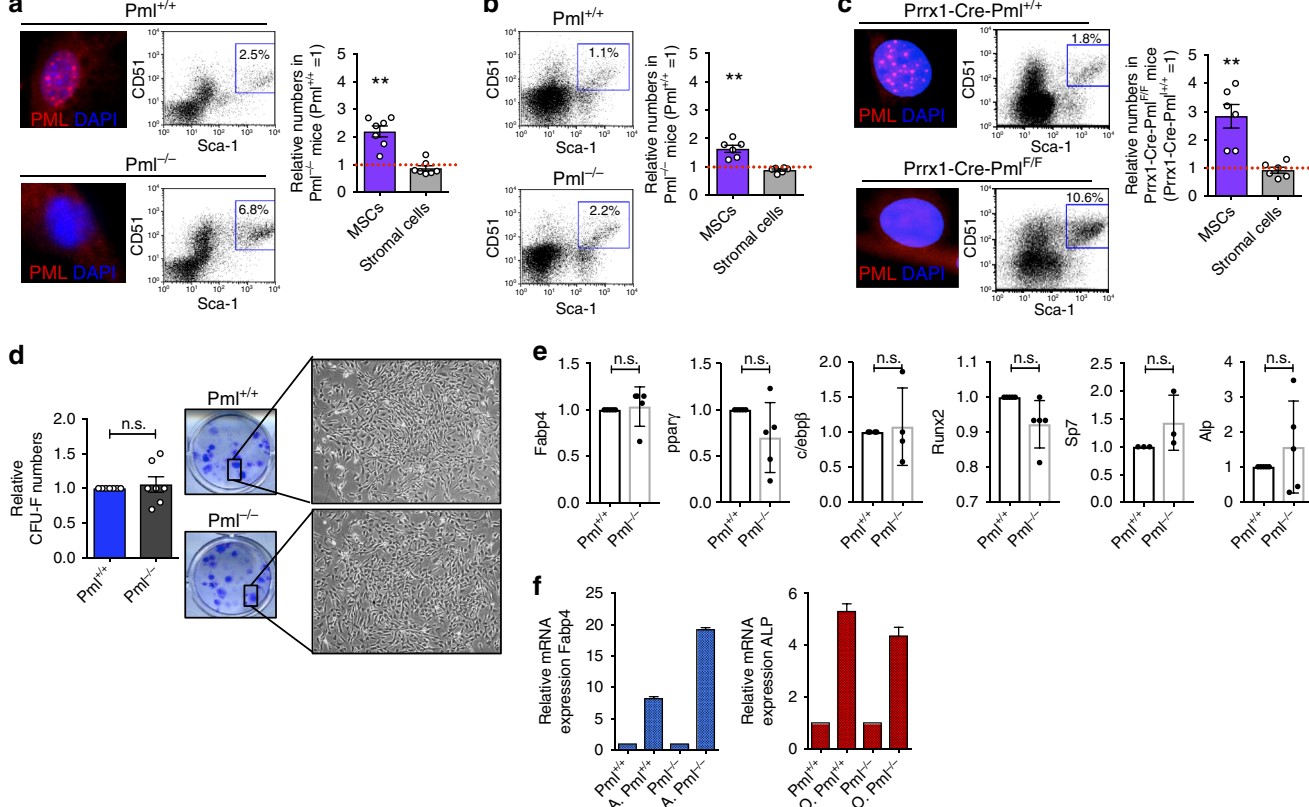

**Fig. 1** Pml regulates MSCs cell autonomously. **a** Pml expression in MSCs derived from $Pml^{+/+}$ or $Pml^{-/-}$ mice is shown on the left. Representative plots in the middle show the percentage of MSCs in $Pml^{+/+}$ or $Pml^{-/-}$ mice, while the chart on the right shows the relative numbers of MSCs, and stromal cells in $Pml^{-/-}$ mice compared to $Pml^{+/+}$ mice ($n = 7$). **b** Percentage and numbers of MSCs in $Pml^{+/+}$ or $Pml^{-/-}$ mice of 15 months of age. The chart on the right shows the relative numbers of MSCs, and stromal cells in $Pml^{-/-}$ mice compared to $Pml^{+/+}$ mice ($n = 6$). **c** Pml expression in MSCs derived from $Prrx1$-$Cre$-$Pml^{+/+}$ or $Prrx1$-$Cre$-$Pml^{F/F}$ mice is shown on the left of each panel. Representative plots in the middle show the percentage of MSCs in $Prrx1$-$Cre$-$Pml^{+/+}$ or $Prrx1$-$Cre$-$Pml^{F/F}$ mice, while the chart on the right shows the relative numbers of MSCs, and stromal cells in $Prrx1$-$Cre$-$Pml^{F/F}$ mice compared to $Prrx1$-$Cre$-$Pml^{+/+}$ mice ($n = 6$). **d** CFU-F colony forming capacity of MSCs derived from $Pml^{+/+}$ or $Pml^{-/-}$ mice after 5 days in hypoxic conditions. The quantification of the colonies is shown on the left, while representative pictures of the colonies stained with crystal violet, and of the morphology of the cells are shown on the right ($n = 7$). **e** Spontaneous differentiation of $Pml^{+/+}$ and $Pml^{-/-}$ MSCs. The expression levels of Pparg, Fabp4, and c/ebpb have been analyzed to determine the spontaneous differentiation of the cells to adipocytes; while the expression levels of Runx2, Sp7, and Alp have been used to investigate the capacity of the cells to differentiate toward osteoblasts. The charts show biological replicates ± SEM. **f** Capacity of $Pml^{+/+}$ and $Pml^{-/-}$ MSCs to differentiate into adipocytes and into osteoblasts once triggered with specific factors. The chart on the left shows the relative expression of Fabp4 in MSCs $Pml^{+/+}$ or $Pml^{-/-}$, cultured in hypoxic conditions for 5 days, and then subjected (or not) to adipogenesis. (A. indicates cells induced to differentiate into adipocytes). The chart on the right shows the relative expression of Alp mRNA in MSCs induced to differentiate to osteoblasts (O indicates cells induced to differentiate into osteoblasts). The charts show one representative experiment, $n = 2$

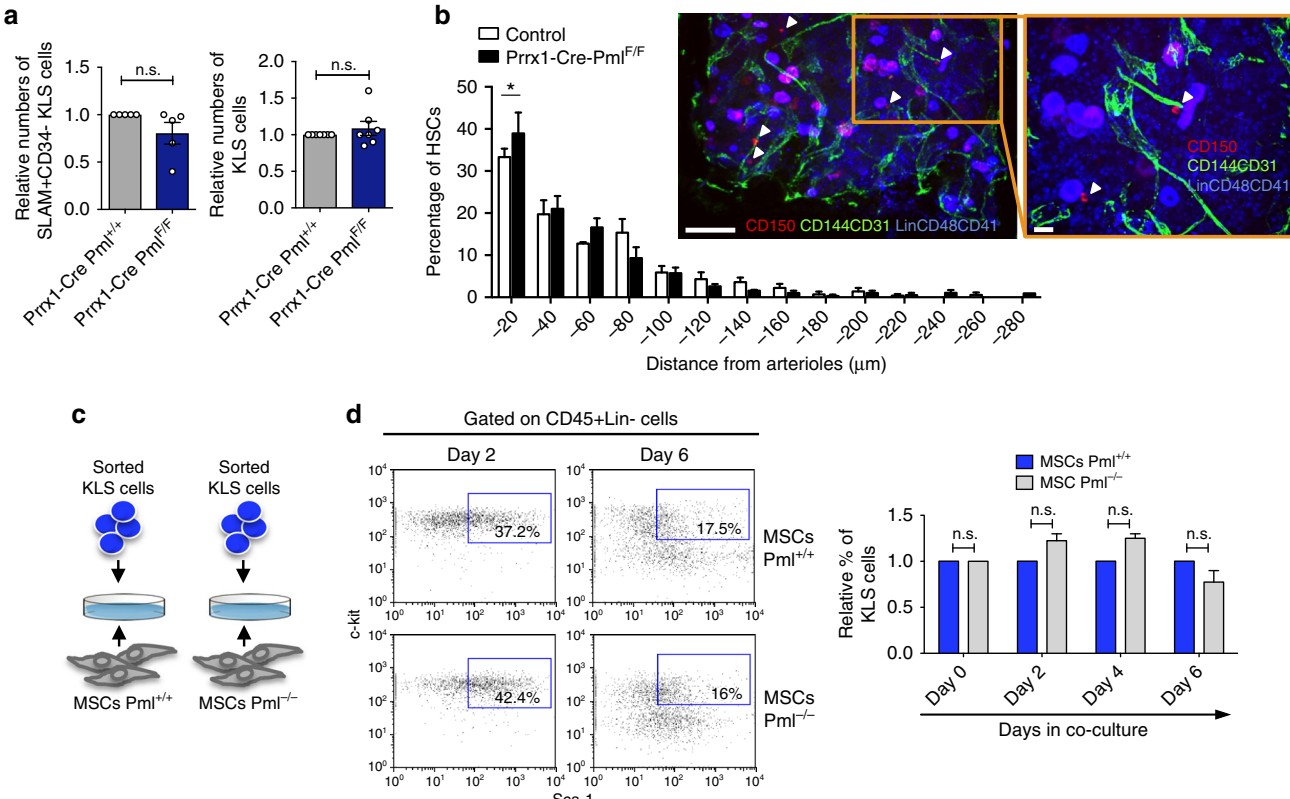

**Fig. 2** Pml regulates only marginally HSCs in a non-cell-autonomous manner. **a** Flow-cytometry analysis of the HSCs compartment of *Prrx1-Cre-Pml*[+/+] and *Prrx1-Cre-Pml*[F/F] mice. SLAM[+]CD34[−]KLS cells are shown on the left panel, while KLS cells are shown in the panel on the right ($n \geq 5$ mice analyzed). Error bars represent the mean $\pm$ SEM. **b** Localization of HSCs relative to arterioles in the sternal bone marrow. $n = 203,219$ HSCs from three mice per control and Prx1-cre[+] group, respectively. Distance between HSCs and arterioles was measured blindly for genotype. Two-sample Kolmogorov–Sminov test; $P = 0.05$. Error bars represent the mean $\pm$ SEM. Representative whole-mount images of sternal bone marrow are shown on the right. Arrowheads denote Lin[−]CD48[−]CD41[−]CD150[+] phenotypic HSCs. Scale bars indicate 100 µm in original image and 20 µm in inset. Vascular endothelial cells are stained with anti-CD31 and anti-CD144 antibodies. **c** Schematic representation of the co-culture assays between MSCs and sorted KLS cells. **d** Co-culture assays between MSCs (*Pml*[+/+] or *Pml*[−/−]) and freshly isolated KLS cells. The representative plots on the left panel show the percentage of KLS cells at day 2 or day 4 of the co-cultures. The chart on the right shows the relative percentage of KLS cells in co-cultures with MSCs *Pml*[+/+] or *Pml*[−/−] at different days. Chart shows one representative experiment $\pm$ SEM out of three independent replicates

*Prrx1-Cre-Pml*[F/F] mice showed slightly higher survival rates, as compared to the control mice (Supplementary Fig. 2c).

**Pml non-cell autonomously regulates the maintenance of leukemic cells.** The observed reduction in the population of BA[+]GFP[+] KLS cells in *Pml*[−/−] mice compared to wild-type recipients led us hypothesize that the maintenance of leukemic cells could depend on the expression of Pml in MSCs, in a non-cell-autonomous manner. In order to test this hypothesis, and to extend this investigation to several types of leukemia, we set up co-culture assays, in which MSCs were cultured together with leukemic cells of different genetic makeup (Fig. 3a and Supplementary Fig. 3a). MSCs were derived from *Pml*[+/+] or *Pml*[−/−] mice. After isolation, MSCs were kept in low oxygen conditions to preserve their stem-cell properties until initiation of the co-cultures[23]. Leukemia of different genetic backgrounds was derived either from transgenic mouse models (*Flt3*[ITD]*IDH2*[R140Q], *BCR/ABL*, and *p53*[−/−] leukemic cells[27,28]) or through transduction of HSCs with retroviral vectors (*HoxA9–Meis1* and *MLL/AF9* leukemic cells) (Supplementary Fig. 3b). Every 4 days, leukemic cells were collected from the previous co-culture, and seeded onto a new layer of MSCs (newly isolated from either *Pml*[−/−] or *Pml*[+/+] mice, and kept in low oxygen conditions). We performed several rounds of co-cultures, and analyzed the total number of leukemic cells present in the first (Supplementary Fig. 3c) and in the

following co-cultures (Fig. 3a). When BCR/ABL, HoxA9–Meis1, and MLL/AF9 leukemic cells were co-cultured with *Pml*[−/−] MSCs, their number was reduced, compared to the co-cultures with *Pml*[+/+] MSCs. Interestingly, no differences were observed when comparing the co-cultures in which *Flt3*[ITD]/*IDH2*[R140Q] or *p53*[−/−] leukemia cells were tested (Fig. 3a). Importantly, when we narrowed our investigation to focus on responsive models, and analyzed the maintenance of the population of LSCs, identified as BA[+]GFP[+]ckit[+]Lin[−]Sca1[+] cells for the CML model (BA[+] KLS cells), and GFP[+]ckit[+] cells for the AML models (AML ckit[+] cells), we observed that the reduction in the number of leukemic cells in the co-cultures with MSCs *Pml*[−/−] correlated with a progressive decline in the number of LSCs cells (Fig. 3b and Supplementary Fig. 3d, e). In order to functionally characterize the leukemic cells within the co-cultures, we focused on the HoxA9–Meis1 and MLL/AF9 models, and we either transplanted the leukemic cells from the co-culture in recipient mice, or we performed methyl-cellulose assays. Accordingly, we found that leukemic cells derived from co-cultures with *Pml*[−/−] MSCs (with both HoxA9–Meis1 and MLL/AF9 models) displayed overt functional disadvantages: leukemic cells showed a marked reduction in their capacity to form colonies when plated as single cells in a methyl-cellulose medium (Supplementary Fig. 4a) and a significantly reduced ability to generate leukemia when transplanted in mice (Fig. 3c and Supplementary Fig. 4b, c).

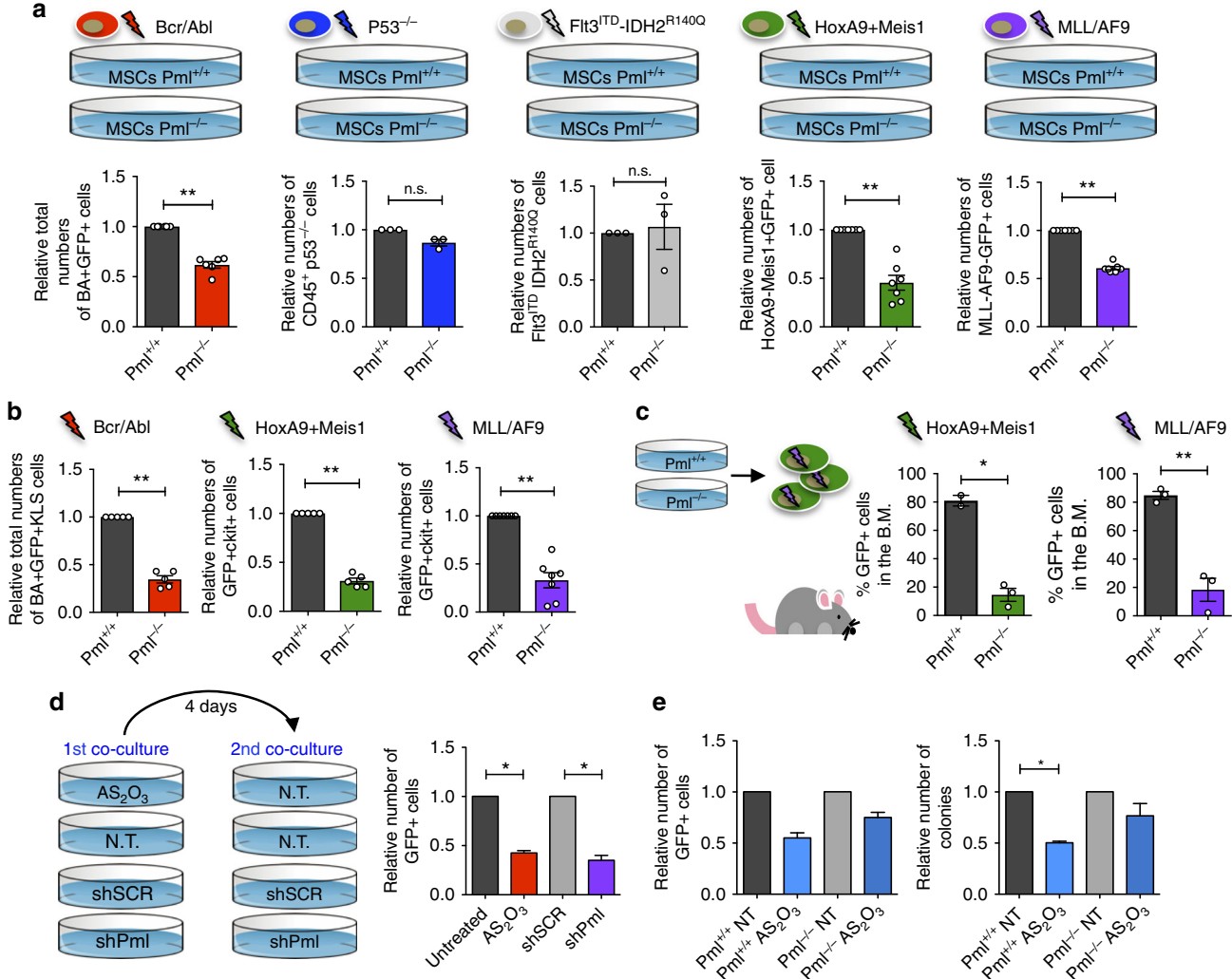

**Fig. 3** Pml regulates leukemic cells in a non-cell-autonomous manner. **a** Relative number of leukemic cells with different genetic background in serial co-culture with MSCs $Pml^{+/+}$ or $Pml^{-/-}$ ($n \geq 3$ biological replicates). Charts represent mean ± SEM. **b** Relative total number of BA[+]GFP[+] KLS cells, GFP[+]ckit[+]Hoxa9[+]Meis1 cells, and GFP[+]ckit[+]MLL/AF9 cells in serial co-culture with MSCs $Pml^{+/+}$ or $Pml^{-/-}$ ($n \geq 5$ biological replicates). Charts represent mean ± SEM. **c** Schematic representation of the experimental design is depicted on the left. Leukemic cells were derived from the last co-cultures and transplanted into recipient wild-type mice. Chart on the left shows the percentage of GFP[+] leukemic cells HoxA9–Meis1 in the bone marrow of recipient mice. Chart on the right shows the percentage of GFP[+] leukemic cells MLL/AF9 in the bone marrow of recipient mice ($n \geq 2$ mice). Charts represent mean ± SEM. **d** Schematic representation of the co-culture assays between MSCs untreated or treated with $AS_2O_3$, or MSCs silenced for the expression of Pml with shRNAs, and leukemic cells (each co-culture was carried on for 4 days). The chart on the right shows the relative numbers of GFP[+] leukemic cells derived from the second co-culture, after the treatment. The chart shows the average of two replicates ± SEM. **e** Relative number of leukemic cells (GFP[+]) in the second co-culture (chart on the left), and relative number of colonies in methyl-cellulose (chart on the right) upon treatment with $AS_2O_3$ of $Pml^{+/+}$ or $Pml^{-/-}$ MSCs. Charts show the average of two replicates ± SEM

Based on these results, we next investigated the potential use of arsenic trioxide ($AS_2O_3$) to impair the maintenance of AML-leukemic cells through MSCs. To this end, we performed co-cultures with MSCs and MLL/AF9 leukemic cells in the presence of $AS_2O_3$. An shRNA against Pml was used as a control in this experiment (Supplementary Fig. 4d). Leukemic cells were collected after treatment with $AS_2O_3$ and plated onto new MSCs in a second round of co-culture, in which the number of leukemic cells was then analyzed (Fig. 3d). We observed that the number of leukemic cells was significantly reduced when the cells were pre-treated with $AS_2O_3$, compared to untreated controls (Fig. 3d and Supplementary Fig. 4e). Importantly, the magnitude of the reduction observed by treating cells with $AS_2O_3$ was on par to that observed by comparing co-cultures performed with MSCs

silenced for Pml or with an shRNA scramble, thus suggesting that $AS_2O_3$ is acting on leukemic cells, at least in part, through reducing the levels of Pml in MSCs. Lastly, in order to address if $AS_2O_3$ could affect the maintenance of leukemic cells in a Pml-independent manner, we decide to employ in these experiments $Pml^{-/-}$ MSCs, and we treated them with $AS_2O_3$, in parallel to wild-type MSCs. As shown in Fig. 3e, while the treatment of wild-type MSCs significantly reduced their capacity to maintain functional leukemic cells, the same treatment showed only minor effects on the $Pml^{-/-}$ MSCs, suggesting that, at least in these settings, $AS_2O_3$ is acting through Pml to sustain leukemic cells.

**The expression of Pml in MSCs contributes to resistance to chemotherapeutic treatment.** Pml contributes to sustaining BA

+KLS cells, and AML ckit+cells in a non-cell-autonomous manner through MSCs. On the basis of these findings, we next investigated whether the blockade of this non-cell-autonomous mechanism could be beneficial in a therapeutic setting. To address this question, we employed in our analysis MLL/AF9[+] GFP[+] leukemic cells, which we injected into $Pml^{-/-}$- or $Pml^{+/+}$- recipient mice. When leukemic cells were detected in the peripheral blood of transplanted mice, we began treatment with Ara-C and doxorubicin, following a regimen that mimics standard induction therapy for patients with daunorubicin[29]. Upon the completion of chemotherapy, we analyzed the persistence of leukemic cells in the BM of mice as a measure of minimal residual disease (Fig. 4a). We observed that treatment with chemotherapy significantly and progressively reduced MLL/AF9[+] GFP[+] leukemic cells in the BM of both $Pml^{-/-}$ and $Pml^{+/+}$ mice, compared to the untreated controls (Fig. 4b). While leukemic blasts were progressively eliminated upon treatment, a subset of leukemic cells including MLL–AF9 GFP[+]ckit[+] cells showed resistance to the treatment and persisted in BM. Importantly, however, MLL–AF9 GFP[+] ckit[+] cells were reduced

in $Pml^{-/-}$ recipients, compared to $Pml^{+/+}$ mice (Fig. 4c), suggesting that the persistence of MLL–AF9 ckit[+] cells after chemotherapy is impaired if the non-cell-autonomous functions of Pml are deficient.

To further demonstrate the possibility of combining chemotherapy treatment with the blockade of the non-cell-autonomous functions of Pml, which sustain leukemic cells, we next utilized $Prrx1$-Cre-$Pml^{F/F}$ mice, and we performed survival experiments. Pml conditional knockout mice were transplanted with MLL/AF9 leukemic cells, subjected to treatment with chemotherapy, and then analyzed in terms of survival rate. By performing this analysis, we observed a significant difference in the survival rate of the two cohorts of recipient mice. $Prrx1$-Cre-$Pml^{F/F}$ recipients displayed a survival advantage after chemotherapy, compared to the controls (Fig. 4d), thus further supporting the hypothesis that the blockade of the non-cell-autonomous mechanisms of maintenance of leukemic cells, which are regulated by Pml, could be beneficial in therapeutic settings.

We next characterized the impact of chemotherapy on leukemic cells in the two cohorts of recipients. To this end, we

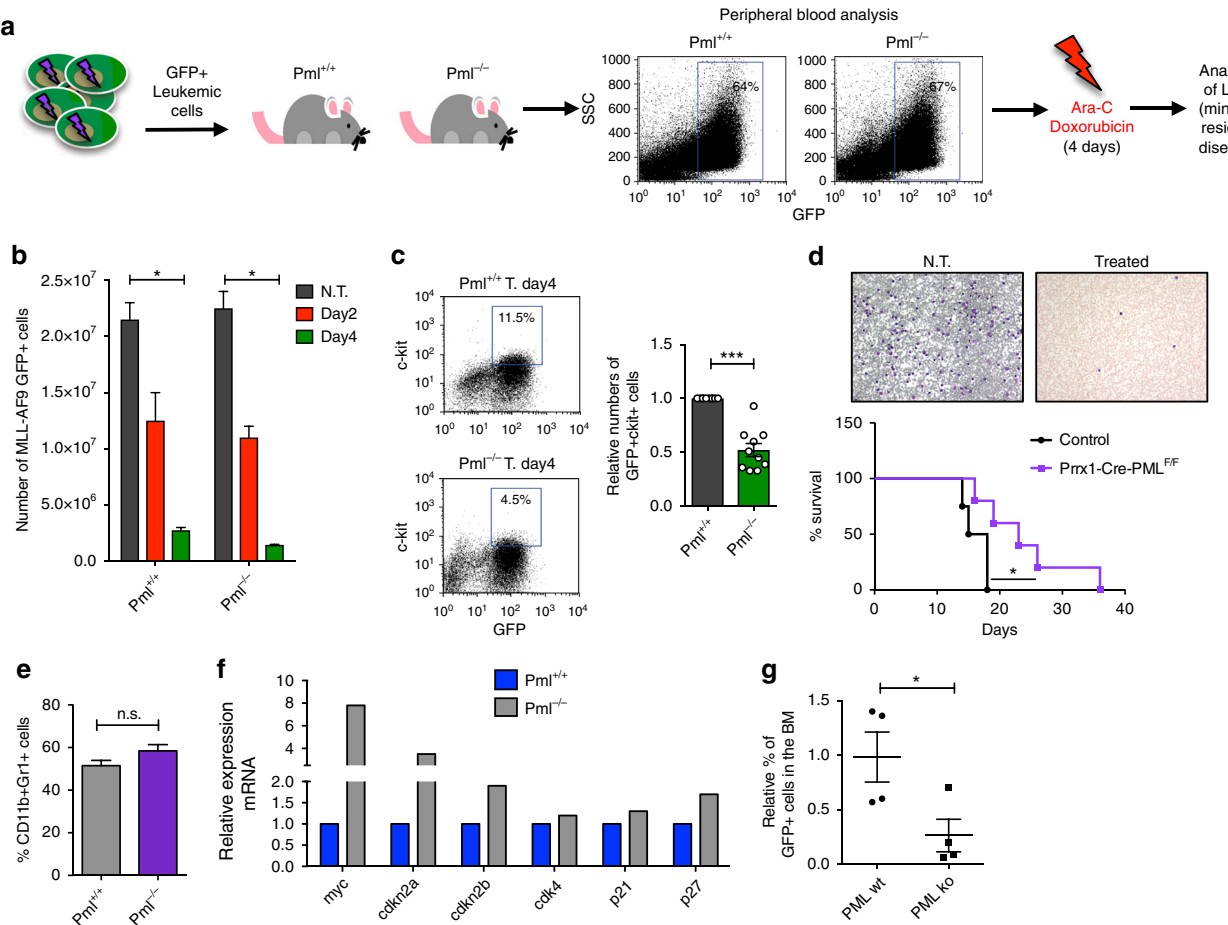

**Fig. 4** Pml sustains non-cell autonomously minimal residual disease in AML upon chemotherapy treatment. **a** Schematic overview of the experimental design for the treatment of $Pml^{+/+}$ or $Pml^{-/-}$ mice with chemotherapeutic drugs. **b** Progressive reduction of leukemic cells upon chemotherapy in treated $Pml^{+/+}$ or $Pml^{-/-}$ mice; $n = 2$ mice for each condition. Charts show average ± SEM. **c** Progressive reduction of percentages (plot on the left) and numbers (chart on the right) of MLL–AF9[+]GFP[+]ckit[+] cells collected from the endosteal bone marrow of $Pml^{+/+}$ or $Pml^{-/-}$ mice treated for 4 days with AraC and doxorubicin ($n = 10$ mice each group analyzed). Charts show the average ± SEM. **d** Survival curve of $Prrx1$-Cre-$Pml^{F/F}$ or control mice that carry MLL/AF9 leukemic cells, and which have been treated with chemotherapy (AraC+Doxorubicin) for 4 days. Leukemic blasts in the blood of treated or not treated mice are shown in the upper panel. **e** Percentage of leukemic cells derived from $Pml^{+/+}$ or $Pml^{-/-}$ mice treated with chemotherapy, which express the surface markers CD11b and Gr1; $n \geq 5$ mice each group. **f** Relative expression of myc, cdkn2a, cdkn2b, cdk4, p21, and p27 in LICs derived from $Pml^{+/+}$ or $Pml^{-/-}$ mice after treatment with chemotherapy (cells of $n = 4$ mice each group have been pooled before RNA extraction). **g** Percentage of GFP[+] leukemic cells in secondary recipient mice transplanted with residual leukemic cells derived from $Pml^{+/+}$ or $Pml^{-/-}$ mice treated with chemotherapy ($n = 4$ mice each group). Data are shown as average ± SEM

analyzed differentiation, apoptosis, and cell cycle profiles of leukemic cells. As shown in Fig. 4e, f, we did not observe differences in terms of expression of differentiation markers (Cd11b and Gr1). However, MLL–AF9 ckit⁺ cells derived from knockout recipients showed higher expression of cell-cycle progression inhibitory genes (Fig. 4f). Accordingly, once cells were collected from the treated animals and then re-transplanted at limiting dilution numbers in secondary recipient mice, the cells that derived from knockout mice showed defects in leukemia reconstitution, when compared to the leukemic cells that were derived from wild-type mice (Fig. 4g).

Taken together, these results corroborate our hypothesis that Pml is able to sustain leukemic cells non-cell autonomously, and suggest that coupling chemotherapy treatment with $AS_2O_3$, or other Pml inhibitors could be beneficial in the treatment of several subgroups of leukemia, including certain types of AML, through the niche.

**Pml expression in MSCs leads to a pro-inflammatory micro-environment**. It has been reported that soluble factors and cytokines are directly involved in the cell-extrinsic maintenance of leukemic cells[11]. Therefore, we tested whether Pml was able to regulate the release of soluble factors and cytokines from MSCs, which could be functionally relevant for the maintenance of leukemic cells. To this end, supernatants of sub-confluent $Pml^{-/-}$ and $Pml^{+/+}$ MSCs were collected and tested by ELISA immunoassays for the presence of more than 90 different soluble factors. This analysis clearly showed that the expression of several pro-inflammatory cytokines was reduced in $Pml^{-/-}$ MSCs, as compared to $Pml^{+/+}$ MSCs (Fig. 5a). By analyzing these factors, we observed a differential expression of Cxcl1 and Il6 between $Pml^{-/-}$ and $Pml^{+/+}$ MSCs. To determine whether Pml could regulate these factors in MSCs at the transcriptional level, we isolated fresh MSCs from wild-type or knockout mice (either total body knock outs or the conditional ones) and we ran RT-qPCR

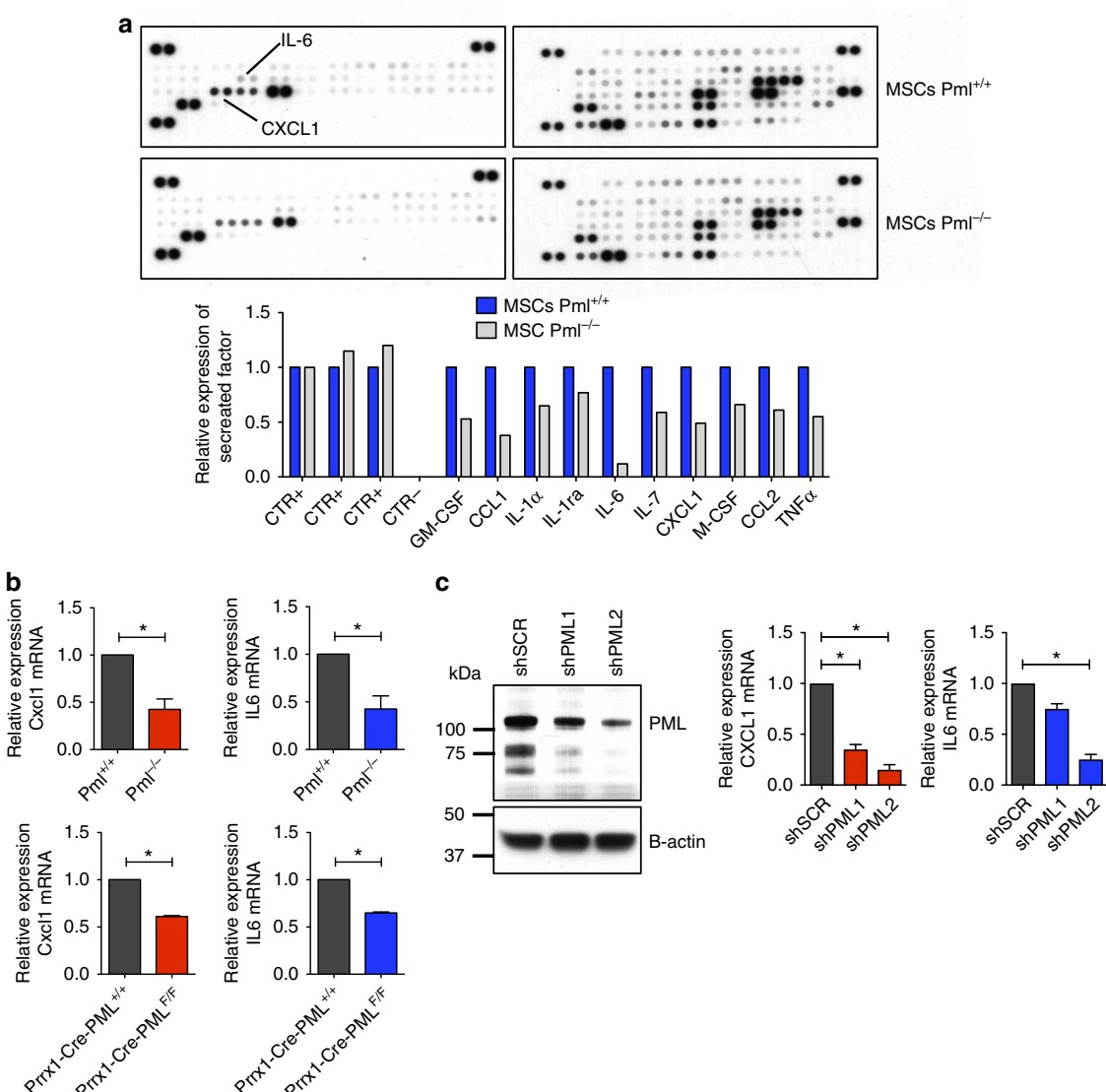

**Fig. 5** Pml regulates pro-inflammatory cytokines in MSCs. **a** ELISA array for the detection of soluble factors and cytokines in the supernatant of MSCs $Pml^{+/+}$ or $Pml^{-/-}$. The quantification of significantly different factors is shown in the lower chart. **b** Relative expression of cxcl1 (charts on the left) and Il6 (charts on the right) in MSCs derived from $Pml^{+/+}$ or $Pml^{-/-}$ mice (upper charts) or from $Prrx1$-$Cre$-$Pml^{+/+}$ and $Prrx1$-$Cre$-$Pml^{F/F}$ mice (charts at the bottom). Cells of $n = 3$ mice each group have been pooled before RNA extraction. Data are shown as average ± SEM. **c** Western blot showing the reduction of Pml expression in HS5 cells treated with two independent shRNAs against Pml is shown on the left; the expression levels of cxcl1 and Il6 in cells silenced for the expression of Pml is shown in the charts on the right. Data are shown as average ± SEM

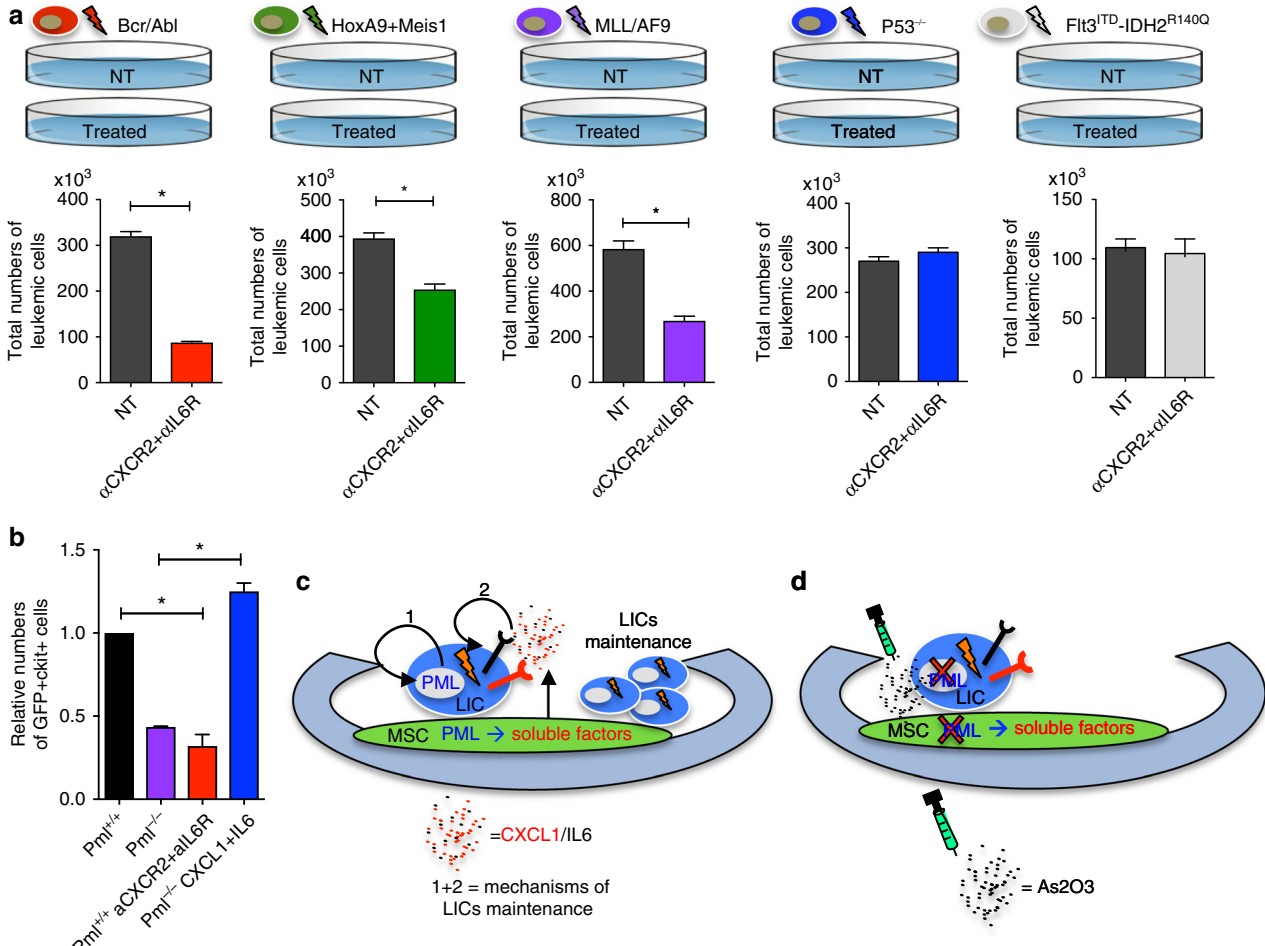

**Fig. 6** Pml regulates LICs non-cell autonomously through Il6 and Cxcl1 pathways. **a** The schematic overview of the experimental design is depicted in the upper panel. The numbers of leukemic cells with different genotype, derived from co-cultures treated with the combination of anti-IL6R and anti-CXCR2, are shown in the lower panel. Data are shown as average ± SEM. (n ≥ 2 independent experiments). **b** Co-cultures of MSCs Pml[+/+] and HoxA9–Meis1[+]GFP[+] leukemic cells were untreated (NT) as control, or treated with anti-CXCR2 in combination of anti-IL6R. Co-cultures with MSCs Pml[−/−] and HoxA9–Meis1[+]GFP[+] leukemic cells were untreated as control, or treated with recombinant Il6 and Cxcl1 proteins. Leukemic cells were then re-plated onto new MSCs (Pml[+/+] or Pml[−/−]); secondary co-cultures were analyzed for the presence of GFP[+]ckit[+] cells. The relative numbers of GFP[+]ckit[+] cells in the different conditions are shown. Data are shown as average ± SEM. **c** Schematic representation of the proposed model. In MSCs, Pml regulates the secretion of pro-inflammatory soluble factors, which sustain the maintenance of leukemic cells and those enriched for LIC capacity. Pml sustains leukemic cells in a non-cell-autonomous manner, in cooperation with the cell-autonomous mechanisms within LICs, as previously described. **d** Pml degradation, (achieved for example through treatment with arsenic trioxide (AS$_2$O$_3$)), contributes to eliminate leukemic cells

analysis. As shown in Fig. 5b, the expression of both Il6 and cxcl1 were significantly reduced in Pml[−/−] cells, compared to those in Pml[+/+]. Moreover, similar results were observed when knocking-down the expression of Pml in HS5 cells, through shRNAs (Fig. 5c).

Our observation led us to hypothesize that Cxcl1/Cxcr2 and Il6/Il6R signals could be involved in the non-cell-autonomous maintenance of leukemic cells that is orchestrated by Pml. To address this important question, we next performed functional assays of serial co-cultures with leukemic cells (both CML and AML) and MSCs, in which the functionality of Cxcr2 and Il6R was blocked with neutralizing antibodies. We set up a serial co-culture protocol in which cells were treated with neutralizing antibodies during the first co-culture; leukemic cells were then collected from these first co-cultures, and re-seeded onto fresh wild-type MSCs for a second co-culture. The number of leukemic cells, and the maintenance of LSCs-enriched population of leukemic cells were then analyzed in both co-cultures (see the schematic representation of the experimental design in Supplementary Fig. 5a). Interestingly, the concomitant blocking of these

signaling pathways was beneficial for Bcr/Abl, HoxA9–Meis1, and MLL/AF9 leukemic cells (Fig. 6a and Supplementary Fig. 6b), while no effects were observed on leukemic cells with a Flt3[ITD]IDH2[R140Q] and a p53[−/−] genetic background (Fig. 6a). Although anti-Cxcr2 or anti-Il6R alone was sufficient to impair the maintenance of BA[+]KLS cells, a concomitant blockade of both Cxcr2 and Il6R resulted in a further reduction of BA[+]KLS cells in the co-cultures; however, treatment with a single neutralizing antibody was not effective for AML models (Supplementary Fig. 5b). For those leukemic models in which the treatment proved to be effective, the sub-population of AML ckit[+] cells was impaired as well (Supplementary Fig. 5c–e). Finally, in order to further validate Cxcl1/Cxcr2 and Il6/Il6R as mechanisms by which Pml regulates leukemic cells non-cell autonomously, we employed the HoxA9–Meis1 AML model, and we performed complementary treatments: we added anti-Cxcr2 and anti-Il6 blocking antibodies to the co-cultures with Pml[+/+] MSCs, while on the contrary we added the recombinant proteins Il6 and Cxcl1 to the co-cultures with Pml[−/−] MSCs. Leukemic cells were then collected from these co-cultures, and seeded in a second round of

co-culture, where the number of AML ckit$^+$ cells was then analyzed. Accordingly, AML ckit$^+$ cells were reduced in the co-cultures with $Pml^{-/-}$ MSCs when untreated, or in the co-cultures with $Pml^{+/+}$ MSCs when treated with anti-Cxcr2 and anti-Il6R, as compared to controls. However, adding back soluble recombinant proteins Cxcl1 and Il6 to the co-culture with $Pml^{-/-}$ MSCs restored the capacity of Pml-deficient MSCs to sustain AML ckit$^+$ cells (Fig. 6d and Supplementary Fig. 5f).

As a whole, these experiments demonstrate that Cxcl1/Cxcr2 and Il6/Il6R are important synergistic signals for the maintenance of leukemic cells, and that Pml is a pivotal up-stream player in the non-cell-autonomous regulation of these signals emanating from MSCs.

## Discussion

In this study, we performed investigations aimed at dissecting the non-cell-autonomous mechanisms involved in the maintenance of leukemic cells harboring different genetic defects. This analysis, allowed us to reach a number of important conclusions:

i.  Pml sustains leukemic cells in a non-cell-autonomous manner through MSCs. We demonstrate that Pml is functional in MSCs. Cells expand in *Pml*-null mice; however, they do not show an impaired capacity to form CFU-F colonies, or to differentiate toward mature cells when triggered to do so. Importantly, however, the presence of Pml in MSCs proves critical for their capacity to sustain leukemic cells, and among them the population of leukemic cells enriched for LSCs, in both in vitro (co-cultures of leukemic cells and MSCs), and in vivo settings. Furthermore, although we observed a significant non-cell-autonomous role for Pml in the maintenance of leukemic cells, Pml's role was marginal in sustaining, non-cell autonomously, normal HSCs.

ii. Leukemic cells with distinctive genetic defects differentially rely on Pml's non-cell-autonomous regulation. Accordingly, leukemic cells carrying the BCR–ABL, MLL–AF9 fusion proteins, and co-expressing HoxA9 and Meis1 were all affected by the inactivation of Pml in MSCs, thus suggesting the possibility that the same therapeutic intervention could be beneficial in multiple leukemia sub-types. Importantly, the burden of minimal residual disease after treatment with conventional chemotherapy was significantly reduced in *Pml*-null recipients compared to controls. These findings further strengthen the hypothesis that inhibitors of Pml (e.g., arsenic trioxide/AS$_2$O$_3$, already in use for the treatment of promyelocytic leukemia) may be used for the treatment of CML, as well as in AML harboring the MLL–AF9 fusion protein or expressing high levels of HoxA9 and Meis1. However, leukemia of varying genetic subtypes differentially relies on the BM stroma[6]. In line with this notion, we observed that $p53^{-/-}$ leukemic cells, and those carrying $Flt3^{ITD}IDH2^{R140Q}$ do not rely on the expression of Pml in MSCs, nor was treatment that blocked cxcr2 and Il6r effective in these models. Thus, our data lend further support to the notion that leukemia treatment should be tailored on the basis of a detailed knowledge of the cross talk between the leukemic cells and the microenvironment, and of the dependencies that this cross-talk in turn elicits.

iii. The expression of Pml in MSCs is essential for the generation of a pro-inflammatory and pro-leukemic micro-environment within the BM. Pro-inflammatory cytokines have been recently described as essential for the maintenance of leukemic cells within a niche in the adipose tissues, and for leukemic cells' evasion of chemotherapy treatment[30]. Here, we report that a pro-inflammatory environment might also be responsible for the maintenance

of the population of leukemic cells enriched for LSCs within the BM niche. Moreover, among the soluble factors regulated by Pml, Cxcl1, and Il6 seem to play an important role in the progression of the leukemia. Cxcl1 and Il6 have been found in the serum of leukemic patients at higher levels compared to controls[31,32], suggesting that, targeting these two specific cytokines among all the others, or their receptors, might be beneficial for patients in conjunction with conventional chemotherapy[33]. In CML, BCR/ABL controls expression of IL6 and, through IL6, establishes paracrine loops that act on the maintenance of leukemic progenitors[11]. However, our data show that while the single blockade of IL6R or CXCR2 could be efficacious in CML, this might not be the case for AML, in which the concomitant inhibition of their signals is critical to target AML ckit$^+$ cells.

Taken together, our results demonstrate that Pml plays an important non-cell-autonomous role in maintaining leukemic cells through the regulation of pro-inflammatory cytokines, including CXCL1 and IL6, from MSCs, in addition to the cell-autonomous functions of Pml in LSC that we previously described[20] (Fig. 6c). Importantly, our findings suggest that the degradation of Pml or the selective inhibition of Pml-induced factors, or even their combination, may be beneficial for the eradication of minimal residual disease in CML and AML (Fig. 6d).

## Methods

**Mice.** MSCs were derived from C57BL/6 wild type, $Pml^{-/-}$ and $Pml^{F/F}Prrx1$-Cre. To generate $Pml^{F/F}$ mice, the Pml genomic sequence was cloned by PCR, and then inserted into the pEZ-LOX-FRT-DT vector. The construct was linearized with BamHI and electroporated into embryonic stem cells. Transfectants were selected in G418 (350 μg/ml) and ganciclovir (2 μM) and expanded for Southern blot analysis. Chimeric mice were produced and then mated with C57BL/6 females (The Jackson Laboratories) (for more details see Chen et al.[34]). Prrx1-Cre mice were purchased from the Jackson Laboratory. For all the in vivo experiments, mice with matched sex and age were used, and were randomly assigned to experimental groups. No animal was excluded from the final experimental analysis; the investigators were not blinded to group allocation. Animal experiments were performed in accordance with the guidelines of Beth Israel Deaconess Medical Center Institutional Animal Care and Use Committee.

**Mesenchymal stem cells maintenance and differentiation.** Long bones (tibia and femurs) were collected, crushed and digested with collagenase II (1 mg/ml) for 1 h and shaken at 37 °C. Recovered cells were stained and FACS sorted as: CD45$^-$CD31$^-$Ter119$^-$Sca1$^+$CD51$^+$, and cultured using complete MesenCult medium (STEMCELL Technologies) and maintained in humidified chamber with 5% CO$_2$ and 1% O$_2$, half medium was changed every 3 days. After 7 days in culture at 1% O$_2$, cells formed visible CFU-F colonies and after this point, cells were split once they reached 80% confluency. When inducing the differentiation, the MesenCult medium was changed with a specific medium for each differentiation as reported in ref. [23]. During the differentiation process, cells were maintained in regular oxygen concentration, 5% CO$_2$ at 37 °C. Mature adipocytes were identified with Oil-Red-O (Sigma) following manufacturer procedures. Mature osteoblasts were stained with Leukocyte Alkaline Phosphatase kit (Sigma) according to the manufacturer procedures.

**Flow cytometry.** Cells were analyzed using LRSII (BD, Pharmingen) and sorted using FACS-ARIA II (BD, Pharmingen). The following antibodies were used: anti-CD45 FITC (clone 30-F11 Biolegend), anti-CD31 FITC (clone MEC13.3 Biolegend), anti-Ter119 FITC (clone TER-119 Biolegend), anti-Sca1 Pacific Blue (clone D7 Biolegend), anti-CD51 PE (clone RMV-7 Biolegend), bio-Lineage panel antibodies [CD4 (clone GK1.5 eBioscience), CD8 (clone 53-6.7 eBioscience), CD3 (clone 145-2C11 eBioscience), Ter119 (clone TER-119 eBioscience), CD11b (clone M1/70 eBioscience), Gr1 (clone RB6-8C5 eBioscience), NK1.1 (clone PK136 eBioscience), B220 (clone RA3-6B2 eBioscience)], anti-ckit APC (clone 2B8 eBioscience), anti-Sca1 PE (clone D7 eBioscience), anti-CD34 FITC (clone RAM-34 eBioscience), anti-SLAM (CD150) PerCP cy5.5 (clone TEC15-12F12.2 Biolegend), anti-Cd11b APC (clone M1/70 Biolegend), and anti-Gr1 PE (clone RB6-8C5 Biolegend).

**Western blots.** For western blot, cell lysates were prepared with RIPA buffer. The following antibodies were used: rabbit anti-PML clone H-238 (Santa Cruz), anti-Pml clone 36.1-104 (Millipore), and anti-β-actin (Sigma-Aldrich). The original blot of Fig. 5c is shown as supplementary Fig. 6.

**Real-time quantitative PCR**. Total RNA was extracted using Triazol reagent (Invitrogen) and retro-transcribed to cDNA using iScript cDNA Synthesis kit (Bio-Rad). For RT-qPCR (Real-time quantitative PCR) performed directly on ex vivo isolated cells, cells were FACS sorted directly into the extraction kit buffer. Pml was detected using TaqMan FAM-conjugated probes (Applied Biosystems). Expression level was normalized to mouse β2-microglobulin.

Fabp4: Mm00445878_m1
ALP: Mm00475834_m1
β2-microglobulin: Mm00437764_m1.

**Immunofluorescence staining**. Cells were fixed with 4% PFA for 10 min, washed with PBS and permeabilized with PBS, Triton X-100 0.2% for 10 min. Blocking before antibodies was performed in PBS, Triton X-100 0.2 and 10% FBS for 30 min. Primary antibody anti-Pml (Millipore) was incubated overnight in blocking buffer, and an anti-Mouse-564 was used as secondary antibody.

**In vitro co-culture assays**. Leukemic cells were generated as described in Figs. 2a and 3a. MSCs were collected from the endosteal surface of the BM as described before, cultured in low oxygen concentration until the formation of CFU-F colonies, and then plated at the same number for the co-cultures. Once MSCs reached 80% confluency, leukemic cells were plated on top of them and the co-cultured started. Leukemic cells were analyzed after 4 days in co-culture. For serial co-cultures, leukemic cells were collected from the previous co-culture, counted and re-plated onto new MSCs after normalization considering the percentage of GFP+ cells. In selected experiments neutralizing antibodies anti-CXCR2 (R&D) (250 ng/ml) and anti-IL6R (Biolegend) (100 ng/ml), or the recombinant proteins CXCL1 and IL6 (both from Biolegend and added at the concentration of 100 ng/ml) were added to the co-cultures. In selected experiments, leukemic cells were derived from the co-cultures and plated in equal number (1000 cells/plate) in the methyl-cellulose. The number of colonies containing more than 50 cells were then counted after 5 days from the plating. In other experiments, total leukemic cells from the co-cultures have been transplanted into wild-type mice. Specifically, we trans-planted 20,000 HoxA9–Meis1 GFP+ leukemic cells, and 1000 or 20,000 MLL/AF9 GFP+ leukemic cells.

**In vivo serial transplantation and chemotherapy treatment**. Leukemic cells were injected in Pml wild type of Pml-null mice in the same number. In total, 50,000 GFP+ cells for the BCR–ABL model, 20,000 GFP+ cells for the MLL–AF9 model, and 50,000 GFP+ cells for the HoxA9–Meis1 model. The development of leukemia was monitored upon periodic bleedings. Once the mice developed leukemia and GFP+ cells were detectable within the blood stream, GFP+ cells were collected from their BM and transplanted in equal numbers into secondary recipients, in the serial transplantation settings, or mice were treated with chemotherapy in selected experiments. Ara-C and Doxorubicin were administrated using the concentrations published in ref. [29]. In selected experiments, residual leukemic cells following treatment with chemotherapy were derived from the BM and used for subsequent experiments. In ex vivo experiments their immune-phenotype and expression profiles were analyzed. Additionally, GFP+ckit+ residual leukemic cells after treatment were sorted, and re-transplanted in Pml wild-type recipients in limiting dilution. The leukemia progression in the recipient mice was followed by periodic bleedings. The percentage of GFP+ leukemic cells in the BM of the recipient mice was then analyzed 2 months after the transplantation.

**ELISA arrays**. To detect soluble protein, MSCs Pml+/+ or Pml−/− were plated at the same numbers, and the supernatant was collected after a few days of culture in hypoxic conditions. Collected supernatants were analyzed within Mouse Cytokine array panel A array kit and Angiogenesis array kit (R&D Systems), following manufacturer instructions.

**Statistical analysis**. No statistical method was used to predetermine sample size. For all statistical analyses, the analysis was done by a two-tailed unpaired Student's $t$ test. Values of $P < 0.05$ were considered statistically significant. *$P < 0.05$; **$P < 0.01$; ***$P < 0.001$ ($t$ test).

**Data availability**. All relevant data are available from the authors.

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

## Acknowledgements

We thank Lauren Southwood, Kaitlyn Doherty, and Elizabeth Stack for their insightful editing, and all members of the Pandolfi lab for critical discussion. Grants Support: Jlenia Guarnerio was supported by a post-doctoral fellowship from American-Italian Cancer Foundation for the execution of this work. This work has been supported by the NIH grants (R01-CA142874 and R35CA197529 to Pier Paolo Pandolfi and R01 DK056638 to P.S.F.).

## Author contributions

J.G. and P.P.P. designed, realized and analyzed the experiments. N.A. and P.F. performed intra-vital microscopy analysis, L.M.M., J.F., K.B., and A.V.M. helped with the experiments. J.G., L.M.M., K.I., P.F. and P.P.P. wrote the manuscript.

## Additional information

**Competing interests:** The authors declare no competing financial interests.

