## [Peer Review File · Nature Communications]

Reviewers' comments:

Reviewer #1 (Remarks to the Author):

Guarnerio and colleagues report a non-cell-autonomous role of PML in maintenance of leukemia-initiating cells (LICs). Authors demonstrate that loss of PML in mesenchymal stem cells (MSCs) results in failure of maintenance of LICs in CML and AML models, but does not affect normal HSCs. Importantly, PML deficiency in MSCs enhanced sensitivity of LICs to chemotherapy, contributing to eradication of residual leukemia against treatment. Mechanistically, authors revealed that PML controls production of CXCL1 and IL6 in MSCs, which support LIC maintenance. This paper is well written and the experiments are well organized. These findings are important in aspect of novel therapeutic approach for leukemia patients. However, present data do not fully support their conclusion. Specific concerns and questions are below.

1. In Fig.1c, Prrx-Cre; Pmlf/f mice show reduction of Pml mRNA in MSCs but not stroma cells. What are exactly "stroma cells"? The stroma cells are not derived from MSCs?
2. In Fig1f, Pml -/- mesenchymal cells proliferate more than Pml+/+ cells. Are the cytokine production failure, such as IL6 and Cxcl1, involved in the MSC phenotype?
3. In Fig.3, authors evaluated numbers and function of LICs in vivo. Authors should add more information about leukemogenesis, e.g., leukemia cell numbers in peripheral blood, pathological analysis of spleen or other tissues, or cell surface marker analysis of leukemia cells.
4. In Fig3k, TKI sensitivity of CML stem cells in Pml mutant mice should be evaluated.
5. Fig4 show that PML is essential for maintenance of LICs in HoxA9/Meis1- and MLL/AF9-driven AML models, but not other models. What are reasons for the difference? Does the difference in responsiveness to cxcl1 or IL6 cause the phenotypes? How are effects of anti-cxcr2/IL6R antibodies on LICs in p53-/- or FLT3/IDH AML?
6. In Fig4f-h, authors showed comparable effects of shPml and AS2O3 on LIC numbers. How is combination effects of shPml (or Pml-/-)and AS2O3 treatment? If AS2O3 has further suppressive effect on LICs on Pml deficient MSCs, AS2O3 may inhibit maintenance of LICs in an independent manner of Pml.
7. In Fig5d, additional information of leukemia phenotypes, not only survival, e.g., leukemia cell numbers in peripheral blood, pathological analysis of spleen or other tissues, or cell surface marker analysis of leukemia cells, should be added. In addition, authors should show mechanisms how LICs are suppressed by Pml deficiency in vivo. Cell cycle, apoptosis and differentiation of LICs should be analyzed.
8. In Fig6, authors showed down-regulation of cytokines, cxcl1 and IL6, in culture media. Does the similar phenomenon occur in vivo? Authors should show cytokine levels in PB or BM plasma in Prrx-Cre; Pmlf/f mice.

Reviewer #2 (Remarks to the Author):

The manuscript by Guarnerio and colleagues studies a hypothesized non-cell-autonomous role for PML expressed in MSC's in sustaining LICs in both CML and AML. Leukemia initiating cells (LIC) are themselves hypothesized to be a source of therapy resistance in leukemia's. PML is a tumor suppressor gene whose protein is localized to nuclear bodies where it may regulate gene expression. Guarneria and colleagues have studied PML null mice (germline) and mice with PML

deleted specifically in MSC's. Both lines have an increase in bone marrow MSC's. PML null MSC's have an increase in in vitro proliferation but similar differentiation capabilities. Interestingly, this increase in MSC's did not affect HSC numbers in vivo. The investigators then study the effect of PML null MSC's on LIC in CML and various AML models. PML null MSC's have a decreased ability to sustain CML LIC and AML LIC in some but not other models. A complex Figure 4 suggests interesting effects but is difficult to interpret as below. The authors go on to suggest that PML dependent MSC effects enhance LIC survival after chemotherapy. Such experiments are often complex and here the effect is modest and incompletely defined (see below). ELISA analysis of supernatants from PML null and + cells demonstrate decreased cytokine production and experiments with IL6 and CXCL1 rescue the co-culture with PML null MSC's. Overall, the authors hypothesize that PML in MSC's regulates cytokine production that regulates LIC numbers (particularly under chemotherapeutic stress) but not HSC's. This is an interesting hypothesis however the data is not clearly presented and some critical experiments are not shown. There is also concern that interesting and obvious experiments are not performed to explore the differences in AML models (i.e. is there a difference in dependency in these models of LIC on cytokines?).

Major concerns:

- 1) Insufficient information is provided to assess the experiment shown in Figure 3j-k. After first transplant, is the number of LSC present in PML + vs. PML null mice equivalent. Phenotypic description (by flow) would help although a functional definition of LIC would be even more informative.
- 2) The difference in survival (Figure 3k) is modest and based on a small number of animals. Was there any qualitative difference in the reasons why the mice died or the phenotype of the BCR/ABL+ disease. A limiting dilution study showing that PML null MSC's decrease CML LIC would greatly enhance the robustness of conclusions here.
- 3) Do the authors have any insight into why PML in MSC's impacts LIC in HoxA9 and MLL/AF9 models but not others. I do not know that phenotype of LIC in each of these models is well defined. In MLL/AF9, not all functional LIC are LSK cells. Is this true for HoxA9 based models? Here, a functional assessment of LIC numbers in each model would enhance the data.
- 4) The important experiment done in Figure 5c again uses phenotype, not functionality to comment on stemness. This experiment should be repeated using a functional assessment of LSC, preferably a LDA experiment into 2ndary recipients.
- 5) Figure 6A. The figure shows a decrease of all cytokines shown in PML null MSC's but the authors choose to focus on IL6 and CXCL1. Was other data confirmed? Why are these two cytokines chosen for study?
- 6) What is the mechanism of the initial expansion of AML and AML LIC's described in Supplemental Figure 3c. This data seems unexpected in the context of other results.
- 7) It would strengthen the manuscript to provide a clearer explanation of how PML regulates cytokine production. Is the mRNA for these cytokines decreased? Does lack of PML generate other markers of a pro-inflammatory state?

Minor concerns:

- 1) Figure 1A/1B – It is not entirely clear how total MSC's was calculated. I assume that some measure of total cells collected was made. Is this from one bone or multiple bones. crushed bone or flushed bone? Please clarify.
- 2) I would prefer that the graph's in Figures 1A and 1B have the same y axis to make comparison to each other clearer.
- 3) Figures 3d-h would be clearer if the passage number was indicated on the figure and graph's.
- 4) The decrease in in vitro cultured BA LSC (Figure 3h) is statistically significant but "dramatic" would seem to over-state the case (c.f. text top of page 9).
- 5) Labels for HoxA9-Meis or MLL/AF9 on Figures 4b-e would increase clarity of data.
- 6) Figure 4d – Are the cells being plated here "generic" leukemic cells or sorted stem cells. This seems to be leukemic cells which does not allow an assessment of the relative functionality of the stem cells. Rather, this data simply confirms Figure 4b-c. Please clarify.

7) How long were cultures shown in Figures 4f-g performed? Was flow done to assess phenotype after first culture?

8) The authors state (p. 12), "Based on their known ability to defy cytotoxic therapy, LICs have been identified as the subpopulation of leukemic cells responsible for the relapse of leukaemia after conventional treatment (REF2). The referenced review does not support this conclusion and, in fact, to my knowledge, this hypothesis has never been experimentally tested (although it is widely assumed to be true).

9) If I am understanding the data correctly, Figure 5c actually shows a decrease in expression of the IL6 receptor when MLL-AF9 cells are cultured on PML negative MSC's. Are the effects seen in Figure 7 due to decreased production of IL6 by MSC's or decreased IL6R on LIC's?

Reviewer #3 (Remarks to the Author):

This manuscript describes a novel role for PML expressed by the mesenchymal stromal cells in the maintenance of leukemia stem cells, elegantly demonstrated in animal models. It is believed that there is appreciable influence of the marrow microenvironment on cell survival in hematologic malignancies, but the specific elements involved and the mechanisms are yet to be fully elucidated. This contribution elegantly illustrates the role of one tumor suppressor gene, PML, on survival of leukemia initiating cells, and reveals a potential therapy that could influence the ability to eradicate the leukemia. Furthermore, the influence of interfering with expression of PML on elucidation of cytokines by the MSCs was investigated and proposed as the mechanism for maintenance of the leukemia initiating cells.

Major recommendations

1. P. 2. It is difficult to understand the sentence, "Here, we report a non-cell-autonomous role for MPL in sustaining LICs in both CML and AML from MSCs." It may be more clear to re-phrase as follows (or similarly): "...role for expression of PML by MSCs in sustaining LICs of both CML and AML."

2. P. 3, line 15, "...ranging from CML, whose cells of origin are hematopoietic stem cells to acute myeloid leukemia which arises from the malignant transformation of hematopoietic progenitors." It might be better to not try to make this distinction, as human CML may arise from a myeloid progenitor as well (Jamieson CH, Ailles LE, Dylla SJ, et al. Granulocyte-macrophage progenitors as candidate leukemic stem cells in blast-crisis CML. *N Engl J Med.* 2004;351(7):657-67).

3. Would the authors wish to address the choice of the term leukemia initiating cells rather than leukemia stem cells or explain how the entities differ?

4. The bolded subtitles refer to PML, but in the text Pml is used. Are the authors distinguishing the PML gene from the Pml protein?

5. P. 12 line 14: "...treatment with Ara-C and doxorubicin, following a regimen that mimics standard induction therapy for patients." Actually, patients with AML receive daunorubicin or idarubicin or mitoxantrone. It is not clear why doxorubicin was chosen instead of daunorubicin, as the latter can be administered to mice as well. If there was a reason for choosing doxorubicin, then a phrase could be added: "...standard induction therapy for patients with doxorubicin substituted for daunorubicin."

6. Regarding Figure 1, for a through d, how were the "stromal cells" isolated in comparison to the MSCs?

7. Regarding Figure 1, it is not possible to appreciate the details of the enlarged regions in part e.

8. For several of the figures, there are so many sub-parts, that it would be better if some of the parts could be moved to supplemental data

9. In the Figure 4 legend, need to define abbreviations in the figures such as SCR as scramble.

Also in this legend, the abbreviation ATO was used but was As2O3 in figure.

Minor comments

1. P. 3. Line 5. Recommend "cell population" rather than "cellular population."

2. P. 4 line 10: "However, the roles of PML in bone marrow microenvironment..." Recommend addition of "the": "...of PML in the bone marrow microenvironment..."
3. P. 10, last line: "a markedly reduction" should be "a marked reduction"
4. P. 21 last paragraph-I believe 50,000....20,000....50,000 are meant rather than 50.000....20.000...50.000....
5. Supplemental figure legend 1: "designed" should be "design"

A non-cell-autonomous role for Pml in the maintenance of leukemia initiating cells from the niche

Reviewers' comments:

Reviewer #1

We thank all the reviewers for their kind words regarding our work, and for the very constructive suggestions, which have greatly helped to strengthen the results. We have addressed their comments in full in the point-by-point rebuttal included here.

Guarnerio and colleagues report a non-cell-autonomous role of PML in maintenance of leukemia-initiating cells (LICs). Authors demonstrate that loss of PML in mesenchymal stem cells (MSCs) results in failure of maintenance of LICs in CML and AML models, but does not affect normal HSCs. Importantly, PML deficiency in MSCs enhanced sensitivity of LICs to chemotherapy, contributing to eradication of residual leukemia against treatment. Mechanistically, authors revealed that PML controls production of CXCL1 and IL6 in MSCs, which support LIC maintenance. This paper is well written and the experiments are well organized. These findings are important in aspect of novel therapeutic approach for leukemia patients. However, present data do not fully support their conclusion. Specific concerns and questions are below.

1. In Fig.1c, Prrx-Cre; Pmlf/f mice show reduction of Pml mRNA in MSCs but not stroma cells. What are exactly "stroma cells"? The stroma cells are not derived from MSCs?

The reviewer's comment is very well taken. We apologize for not having provided sufficient details about what we have defined as "stromal cells" and "MSCs". We labeled "stromal cells" as the population of cells that are derived from the digestion of bone fragments, and which do not express the following surface markers: CD45 (specific marker for hematopoietic cells), CD31 (specific marker of endothelial cells), Ter119 (marker for erythroid precursors), CD51/PDGFR α and Sca1. Because this population is not characterized by the expression of any specific surface markers, its composition is likely heterogeneous, and therefore in need of further investigation. We do know, however, that when isolated and cultured, some cells belonging to this population can assume a fibroblastic-like shape *in vitro* (Guarnerio et. al., Stem Cell Reports 2014); for this reason, we decided to label the cells as "stromal cells". We understand how this labeling could create some confusion for readers. Thus, in order to avoid any misinterpretation, we have now clarified the immunophenotype of the cells analyzed (Supplementary Figure 1b in the revised manuscript), and we labeled this specific population of cells as "stroma CD51-Sca1-", instead of generically as "stromal cells".

2. In Fig1f, Pml -/- mesenchymal cells proliferate more than Pml+/+ cells. Are the cytokine production failure, such as IL6 and Cxcl1, involved in the MSC phenotype?

It is not likely that the differential release of Il6 and Cxcl1 by the two groups of MSCs is responsible for their differential proliferation rate. Indeed, while the differential production of Il6 and Cxcl1 can be detected *in vitro* at very early passages, the cells acquire differential proliferation rate only if they are kept in culture for several passages (see Supplementary Figure 1f of the revised manuscript). This temporal gap between the two events suggests, in our opinion,

that other mechanisms and signaling might be responsible for the proliferation rate of the MSCs. Accordingly, *Pml* has already been described as a regulator of both senescence and terminal differentiation in the cells. In this respect, the increased proliferation rate of *Pml*^{-/-} cells might be due to a bypass of the senescence in culture, which might on the contrary affect the wild type MSCs.

3. In Fig.3, authors evaluated numbers and function of LICs in vivo. Authors should add more information about leukemogenesis, e.g., leukemia cell numbers in peripheral blood, pathological analysis of spleen or other tissues, or cell surface marker analysis of leukemia cells.

As requested, we have now focused more deeply on the AML models of leukemia and provided much more details about the AML-LICs. See new Figure 4 of this revised manuscript.

4. In Fig3k, TKI sensitivity of CML stem cells in *Pml* mutant mice should be evaluated.

In the current manuscript, we studied the non-cell-autonomous role of *Pml* in sustaining leukemic cells using only *Pml*-null mice, either total body or conditional knock-outs. Likely the reviewer refers to “*Pml* null mice” as *Pml*-mutants. If this is the case, as we are sure, by using a CML model, we have performed similar experiments to the one performed in the MLL/AF9 model. To this end, we treated mice with AraC+Doxorubicin, and observed in a CML model results similar to those obtained with the AML model (See Figure R1). Because the work included here focuses on the AML models, we are not showing this experiment in the current manuscript, unless the reviewer deems it appropriate.

Guarnerio et al. Fig. R1

5. Fig4 show that PML is essential for maintenance of LICs in HoxA9/Meis1- and MLL/AF9-driven AML models, but not other models. What are reasons for the difference? Does the difference in responsiveness to cxcl1 or IL6 cause the phenotypes? How are effects of anti-cxcr2/IL6R antibodies on LICs in p53^{-/-} or FLT3/IDH AML?

This is an important question, and we thank the reviewer for raising this point. In order to address this question, we have now treated leukemic cells, with a p53^{-/-} genetic background or FLT3/IDH2mut, with anti-cxcr2/IL6R antibodies, following the same experimental design as for the Bcr/Abl, MLL/AF9 and Hoxa9-Meis1 models. We have then evaluated the responsiveness of leukemic cells to the treatment in terms of total number of leukemic cells left in the co-cultures. As presented in Figure 6 of our revised manuscript, the treatment was ineffective for the leukemic cells from p53^{-/-} or FLT3/IDH mutants, suggesting that these genetic makeups might be not sensitive to signals transduced by Il6 and Cxcl1, and therefore to the non-cell-autonomous regulation orchestrated by Pml. However, because LICs in these AML models (p53^{-/-} or FLT3/IDH) are not well characterized, the analysis of the specific sub-population of LICs upon treatment would be very cumbersome.

6. In Fig4f-h, authors showed comparable effects of shPml and AS2O3 on LIC numbers. How is combination effects of shPml (or Pml^{-/-}) and AS2O3 treatment? If AS2O3 has further suppressive effect on LICs on Pml deficient MSCs, AS2O3 may inhibit maintenance of LICs in an independent manner of Pml.

We thank the reviewer for raising this important point. In order to address it, we treated *Pml*^{-/-} cells, in parallel to the *Pml*^{+/+} cells, with As₂O₃. As shown in Figure 3g of our revised manuscript, while the treatment of wild type MSCs significantly reduced their capacity to maintain functional leukemic cells, the same treatment showed only minor effects for the co-cultures with *Pml*^{-/-} MSCs. We believe that this is an important addition to the study and we thank the reviewer for this suggestion.

7. In Fig5d, additional information of leukemia phenotypes, not only survival, e.g., leukemia cell numbers in peripheral blood, pathological analysis of spleen or other tissues, or cell surface marker analysis of leukemia cells, should be added. In addition, authors should show mechanisms how LICs are suppressed by Pml deficiency *in vivo*. Cell cycle, apoptosis and differentiation of LICs should be analyzed.

We thank the reviewer for this thoughtful feedback. We have further analyzed the AML leukemia model upon treatment with chemotherapy, and dissected the possible *in vivo* mechanisms behind the suppression of LICs upon Pml deficiency. A better characterization of the model, along with new data has been added to the Figure 4 of the revised manuscript.

Specifically in this new figure we show:

- 1- The analysis of the differentiation status (expression of CD11b and Gr1) of the leukemic cells from *Pml*^{+/+} or *Pml*^{-/-} recipient mice upon chemotherapy (Figure 4e of the revised manuscript).
- 2- The expression of cell-cycle regulators in LICs sorted from *Pml*^{+/+} or *Pml*^{-/-} recipient mice treated with chemotherapy (Figure 4f of the revised manuscript).
- 3- Limiting dilution assay in secondary recipients performed with c-kit⁺ LICs derived from the *Pml*^{+/+} or *Pml*^{-/-} recipient mice treated with chemotherapy (Figure 4g of the revised manuscript).

8. In Fig6, authors showed down-regulation of cytokines, cxcl1 and IL6, in culture media. Does the similar phenomenon occur *in vivo*? Authors should show cytokine levels in PB or BM plasma in Prrx-Cre; Pmlf/f mice.

Regarding the experiment suggested by the reviewer, we fear that, since IL6 and CXCL1 are pro-inflammatory cytokines that can be released by cells of any organ in the body, including other

cells present into the bone marrow (e.g. hematopoietic cells or cells of the immune-system), the levels of these proteins in sera from mice would not be representative of what is happening in the bone marrow, and at the level of the cross-talk interaction between MSCs and leukemic cells. Thus, in order address the suggestion of the reviewer and to better define the role of Pml in regulating these soluble factors *in vivo*, we analyzed the expression of these cytokines in MSCs directly isolated from mice. We isolated MSCs from wild type mice or from knock out mice, either total body or conditional knock-outs, and we analyzed the expression of the cytokines by RT-qPCR. We found that Pml regulates the expression of cxcl1 and Il6 at the transcriptional level. These results further confirm that the expression of both cytokines is reduced in *Pml*^{-/-} cells, compared to wild type cells. Moreover, in order to investigate whether the Pml-mediated transcriptional regulation of cxcl1 and Il6 would also occur in human cells, we used 2 different shRNAs to knock down PML in HS5, and we once again performed RT-qPCR analysis of the expression of IL6 and CXCL1. Also in this setting, the absence of Pml significantly impaired the expression of these two cytokines. These new results are shown in Figure 5 of our revised manuscript.

Reviewer #2

We thank all the reviewers for their kind words regarding our work, and for the very constructive suggestions, which have greatly helped to strengthen the results. We have addressed their comments in full in the point-by-point rebuttal included here.

The manuscript by Guarnerio and colleagues studies a hypothesized non-cell-autonomous role for PML expressed in MSC's in sustaining LICs in both CML and AML. Leukemia initiating cells (LIC) are themselves hypothesized to be a source of therapy resistance in leukemia's. PML is a tumor suppressor gene whose protein is localized to nuclear bodies where it may regulate gene expression. Guarneria and colleagues have studied PML null mice (germline) and mice with PML deleted specifically in MSC's. Both lines have an increase in bone marrow MSC's. PML null MSC's have an increase in in vitro proliferation but similar differentiation capabilities. Interestingly, this increase in MSC's did not affect HSC numbers in vivo. The investigators then study the effect of PML null MSC's on LIC in CML and various AML models. PML null MSC's have a decreased ability to sustain CML LIC and AML LIC in some but not other models. A complex Figure 4 suggests interesting effects but is difficult to interpret as below. The authors go on to suggest that PML dependent MSC effects enhance LIC survival after chemotherapy. Such experiments are often complex and here the effect is modest and incompletely defined (see below). ELISA analysis of supernatants from PML null and + cells demonstrate decreased cytokine production and experiments with IL6 and CXCL1 rescue the co-culture with PML null MSC's. Overall, the authors hypothesize that PML in MSC's regulates cytokine production that regulates LIC numbers (particularly under chemotherapeutic stress) but not HSC's. This is an interesting hypothesis however the data is not clearly presented and some critical experiments are not shown. There is also concern that interesting and obvious experiments are not performed to explore the differences in AML models (i.e. is there a difference in dependency in these models of LIC on cytokines?).

Major concerns:

- 1) Insufficient information is provided to assess the experiment shown in Figure 3j-k. After first transplant, is the number of LSC present in PML + vs. PML null mice equivalent. Phenotypic description (by flow) would help although a functional definition of LIC would be even more informative.
- 2) The difference in survival (Figure 3k) is modest and based on a small number of animals. Was there any qualitative difference in the reasons why the mice died or the phenotype of the BCR/ABL+ disease. A limiting dilution study showing that PML null MSC's decrease CML LIC would greatly enhance the robustness of conclusions here.

We completely agree with the reviewer that limiting dilution assays would strengthen our conclusions. Following the reviewer suggestion, we have now focused our study more deeply on the AML model of leukemia, and performed limiting dilution experiments. These results are shown in Figure 4 of this revised manuscript, along with a better characterization of the AML-LICs. Moreover, because the knockout of Pml within MSCs affects LICs, and those leukemic cells that strictly depend on MSCs for their maintenance, a combinatorial approach that couples the blockade of Pml, together with chemotherapy treatment could be ideal in this scenario. Accordingly, similarly to what we have reported for the AML model (Figure 4 of the revised manuscript) we have tested this combinatorial approach on CML leukemia. In this experiment we treated mice with AraC+Doxorubicin, and we obtained, for CML, similar results to those obtained with the AML model (See Figure R1). These results demonstrated a marked difference of CML leukemic cells in Pml+/+ and Pml-/- recipient mice upon chemotherapy. The LICs

population is reduced in the Pml-null recipients compared to the wild type controls, in accordance to what observed in the serial transplantation experiment shown in supplementary Fig 2b of this revised manuscript. Because the study focuses mainly on the AML models, we are only showing this new figure for the reviewer's perusal, unless the reviewer deems appropriate that we add to the supplementary information.

Guarnerio et al. Fig. R1

3) Do the authors have any insight into why PML in MSC's impacts LIC in HoxA9 and MLL/AF9 models but not others. I do not know that phenotype of LIC in each of these models is well defined. In MLL/AF9, not all functional LIC are LSK cells. Is this true for HoxA9 based models? Here, a functional assessment of LIC numbers in each model would enhance the data.

This is an important question, and we thank the reviewer for raising this point. In order to address this question, we have now treated leukemic cells, with a p53^{-/-} genetic background or FLT3/IDH2mut, with anti-cxcr2/IL6R antibodies, following the same experimental design as for the Bcr/Abl, MLL/AF9 and Hoxa9-Meis1 models. We have then evaluated the responsiveness of leukemic cells to the treatment in terms of total number of leukemic cells left in the co-cultures. As presented in Figure 6 of our revised manuscript, the treatment was ineffective for the leukemic cells from p53^{-/-} or FLT3/IDH mutants, suggesting that these genetic makeups might be not sensitive to signals transduced by Il6 and Cxcl1, and therefore to the non-cell-autonomous regulation orchestrated by Pml. However, because LICs in these AML models (p53^{-/-} or FLT3/IDH) are not well characterized, the analysis of the specific sub-population of LICs upon treatment would be very cumbersome.

4) The important experiment done in Figure 5c again uses phenotype, not functionality to comment on stemness. This experiment should be repeated using a functional assessment of LSC, preferably a LDA experiment into 2ndary recipients.

We thank the reviewer for this thoughtful feedback. We have further analyzed the AML leukemia model upon treatment with chemotherapy, and dissected the possible *in vivo* mechanisms behind

the suppression of LICs upon Pml deficiency. A better characterization of the model, along with new data has been added to the Figure 4 of the revised manuscript.

Specifically in this new figure we show:

- 1- The analysis of the differentiation status (expression of CD11b and Gr1) of the leukemic cells from *Pml*^{+/+} or *Pml*^{-/-} recipient mice upon chemotherapy (Figure 4e of the revised manuscript).
- 2- The expression of cell-cycle regulators in LICs sorted from *Pml*^{+/+} or *Pml*^{-/-} recipient mice treated with chemotherapy (Figure 4f of the revised manuscript).
- 3- Limiting dilution assay in secondary recipients performed with c-kit⁺ LICs derived from the *Pml*^{+/+} or *Pml*^{-/-} recipient mice treated with chemotherapy (Figure 4g of the revised manuscript).

5) Figure 6A. The figure shows a decrease of all cytokines shown in PML null MSC's but the authors choose to focus on IL6 and CXCL1. Was other data confirmed? Why are these two cytokines chosen for study?

We chose to focus our attention specifically on these two cytokines because clinical studies suggest the relevance of these cytokines in acute leukemia (See in the manuscript Refs 34-35). Nevertheless, the reviewer comment is well taken, and we cannot exclude the fact that other soluble factors might play a role.

6) What is the mechanism of the initial expansion of AML and AML LIC's described in Supplemental Figure 3c. This data seems unexpected in the context of other results.

We thank the reviewer for this comment. As shown in Supplementary Figure 3c of the revised manuscript, we observed a slight initial expansion of the LICs in the first co-cultures of leukemic cells with *Pml*^{-/-} MSCs, compared to *Pml*^{+/+} MSCs. Interestingly, a similar behavior was also observed for *Pml*^{-/-} HSCs and *Pml*^{-/-} CML cells. In this context, the maintenance of the stem cells was impaired after an initial boost and increased expansion of *Pml*^{-/-} hematopoietic cells, compared to the wild type ones (Ito K, Nature 2008). It is known that forced exit from quiescence often results in symmetric commitment. Indeed, this was the case in a cell autonomous *Pml*^{-/-} HSC-CML model (Ito K., Nature Medicine 2012). Thus it is tempting to speculate that the same is true when *Pml* is inactivated in MSCs, whereby exit from quiescence is followed by commitment towards differentiation.

7) It would strengthen the manuscript to provide a clearer explanation of how PML regulates cytokine production. Is the mRNA for these cytokines decreased? Does lack of PML generate other markers of a pro-inflammatory state?

We agree with the reviewer that understanding how Pml can regulate the cytokine production would strengthen our manuscript. Accordingly, we have performed a RT-qPCR analysis of MSCs *Pml*^{+/+} and *Pml*^{-/-}, and we observed that a marked down-regulation of Il6 and Cxcl1 levels of RNA in *Pml*^{-/-} MSCs. Importantly, we observed this transcriptional regulation both in freshly sorted MSCs and in the MSCs kept in culture. Moreover, a similar down-regulation of Il6 and Cxcl1 was observed also in HS5 cells, upon the knock-down of Pml through 2 independent shRNAs. All these results are shown in Figure 5 of this revised manuscript.

Minor concerns:

1) Figure 1A/1B – It is not entirely clear how total MSC's was calculated. I assume that some measure of total cells collected was made. Is this from one bone or multiple bones. Crushed bone or flushed bone? Please clarify.

The total number of MSCs was calculated based on the number of total stromal cells that we collected from bones. For each mouse, femurs and the tibiae were pooled together before bone crushing. We have clarified these points in the method section.

2) I would prefer that the graph's in Figures 1A and 1B have the same y axis to make comparison to each other clearer.

Following the reviewer suggestion, graphs with the same y axis are now shown in this revised manuscript.

3) Figures 3d-h would be clearer if the passage number was indicated on the figure and graph's.

We have added this information to the figures showing co-culture experiments.

4) The decrease in in vitro cultured BA LSC (Figure 3h) is statistically significant but “dramatic” would seem to over-state the case (c.f. text top of page 9).

Following the reviewer suggestion, we removed this statement from the text.

5) Labels for HoxA9-Meis or MLL/AF9 on Figures 4b-e would increase clarity of data.

We thank the reviewer for this suggestion. These labels have now been added in the figures.

6) Figure 4d – Are the cells being plated here “generic” leukemic cells or sorted stem cells. This seems to be leukemic cells which does not allow an assessment of the relative functionality of the stem cells. Rather, this data simply confirms Figure 4b-c. Please clarify.

We thank the reviewer for this suggestion. We have now clarified this point in the method section.

7) How long were cultures shown in Figures 4f-g performed? Was flow done to assess phenotype after first culture?

Each co-culture was carried out for 4 days. We have added this information in the figure legend as well as within the methods section. Based on the results shown in Figure 3a and Supplementary Figure 3c-d of this revised manuscript, we decide to analyze the second co-cultures also for the experiments shown in the Figures 3d-e.

8) The authors state (p. 12), “Based on their known ability to defy cytotoxic therapy, LICs have been identified as the subpopulation of leukemic cells responsible for the relapse of leukaemia after conventional treatment (REF2). The referenced review does not support this conclusion and,

in fact, to my knowledge, this hypothesis has never been experimentally tested (although it is widely assumed to be true).

We thank the reviewer for pointing out this issue. We have adjusted the text accordingly.

9) If I am understanding the data correctly, Figure 5c actually shows a decrease in expression of the IL6 receptor when MLL-AF9 cells are cultured on PML negative MSC's. Are the effects seen in Figure 7 due to decreased production of IL6 by MSC's or decreased IL6R on LIC's?

Our data suggest that the expression of Il6 is impaired if Pml is lost in MSCs. We have not characterized the expression levels of IL6R in this context because we focused on MSCs direct effects. In order to avoid misunderstandings, we have now changed the Figure 4c in this revised manuscript.

Reviewer #3

We thank all the reviewers for their kind words regarding our work, and for the very constructive suggestions, which have greatly helped to strengthen the results. We have addressed their comments in full in the point-by-point rebuttal included here.

This manuscript describes a novel role for PML expressed by the mesenchymal stromal cells in the maintenance of leukemia stem cells, elegantly demonstrated in animal models. It is believed that there is appreciable influence of the marrow microenvironment on cell survival in hematologic malignancies, but the specific elements involved and the mechanisms are yet to be fully elucidated. This contribution elegantly illustrates the role of one tumor suppressor gene, PML, on survival of leukemia initiating cells, and reveals a potential therapy that could influence the ability to eradicate the leukemia. Furthermore, the influence of interfering with expression of PML on elucidation of cytokines by the MSCs was investigated and proposed as the mechanism for maintenance of the leukemia initiating cells.

Major recommendations

1. P. 2. It is difficult to understand the sentence, “Here, we report a non-cell-autonomous role for MPL in sustaining LICs in both CML and AML from MSCs.” It may be more clear to re-phrase as follows (or similarly): “...role for expression of PML by MSCs in sustaining LICs of both CML and AML.”

We thank the reviewer for this suggestion, and we have re-phrased the sentence accordingly.

2. P. 3, line 15, “...ranging from CML, whose cells of origin are hematopoietic stem cells to acute myeloid leukemia which arises from the malignant transformation of hematopoietic progenitors.” It might be better to not try to make this distinction, as human CML may arise from a myeloid progenitor as well (Jamieson CH, Ailles LE, Dylla SJ, et al. Granulocyte-macrophage progenitors as candidate leukemic stem cells in blast-crisis CML. N Engl J Med. 2004;351(7):657–67).

We agree with the reviewer, and we have changed the text accordingly.

3. Would the authors wish to address the choice of the term leukemia initiating cells rather than leukemia stem cells or explain how the entities differ?

As the reviewer knows, the terms Leukemia Stem Cells (LSC) or Leukemia Initiating Cells (LIC) have been used rather interchangeably in the past years to describe a population of cancer cells, which display cancer regenerative potential, hence at the basis of clinical relapse. We just decided to use one of the two terms. However, we understand that this choice could create some confusion. Therefore, in order to clarify, we have indicated both terms in the introduction of this revised manuscript.

4. The bolded subtitles refer to PML, but in the text Pml is used. Are the authors distinguishing the PML gene from the Pml protein?

We have edited the text so the use of “Pml” is consistent throughout.

5. P. 12 line 14: "...treatment with Ara-C and doxorubicin, following a regimen that mimics standard induction therapy for patients." Actually, patients with AML receive daunorubicin or idarubicin or mitoxantrone. It is not clear why doxorubicin was chosen instead of daunorubicin, as the latter can be administered to mice as well. If there was a reason for choosing doxorubicin, then a phrase could be added: "...standard induction therapy for patients with doxorubicin substituted for daunorubicin."

We agree with the reviewer, and have thus changed the text accordingly.

6. Regarding Figure 1, for a through d, how were the "stromal cells" isolated in comparison to the MSCs?

The reviewer's comment is very well taken. We apologize for not having provided sufficient details about what we have defined as "stromal cells" and "MSCs". We labeled "stromal cells" as the population of cells that are derived from the digestion of bone fragments, and which do not express the following surface markers: CD45 (specific marker for hematopoietic cells), CD31 (specific marker of endothelial cells), Ter119 (marker for erythroid precursors), CD51/PDGFR α and Sca1. Because this population is not characterized by the expression of any specific surface markers, its composition is likely heterogeneous, and therefore in need of further investigation. We do know, however, that when isolated and cultured, some cells belonging to this population can assume a fibroblastic-like shape *in vitro* (Guarnerio et. al., Stem Cell Reports 2014); for this reason, we decided to label the cells as "stromal cells". We understand how this labeling could create some confusion for readers. Thus, in order to avoid any misinterpretation, we have now clarified the immunophenotype of the cells analyzed (Supplementary Figure 1b in the revised manuscript), and we labeled this specific population of cells as "stroma CD51-Sca1-", instead of generically as "stromal cells".

7. Regarding Figure 1, it is not possible to appreciate the details of the enlarged regions in part e.

We have now included higher-quality pictures in Figure 1d of the revised manuscript.

8. For several of the figures, there are so many sub-parts, that it would be better if some of the parts could be moved to supplemental data

We thank the reviewer for this suggestion. On this basis, we have re-formatted the figures in order to show only the key data in the main figures. The additional figures have been moved to the supplementary figures.

9. In the Figure 4 legend, need to define abbreviations in the figures such as SCR as scramble. Also in this legend, the abbreviation ATO was used but was As2O3 in figure.

We have adjusted the text associated with figure 4 as well as the legend to the figure.

Minor comments

1. P. 3. Line 5. Recommend "cell population" rather than "cellular population."

2. P. 4 line 10: “However, the roles of PML in bone marrow microenvironment...” Recommend addition of “the”: “...of PML in the bone marrow microenvironment....”

3. P. 10, last line: “a markedly reduction” should be “a marked reduction”

4. P. 21 last paragraph-I believe 50,000....20,000....50,000 are meant rather than 50.000....20.000...50.000....

5. Supplemental figure legend 1: “designed” should be “design”

We really thank the reviewer for bringing up these points. We have now corrected these errors in both text and figure legends.

Reviewers' comments:

Reviewer #1 (Remarks to the Author):

Authors properly addressed all my questions.

Reviewer #2 (Remarks to the Author):

This is a revised manuscript by Guarnerio and colleagues studying the hypothesis of a non-cell-autonomous role for PML expressed in MSC's in sustaining LICs in both CML and AML. I thank the authors for their revisions and for addressing my concerns. To again summarize, the authors have studied PML null and mice with Cre mediated deletion of PML in the MSC compartment. They demonstrate that loss of PML in MSC leads to an increase in MSC number without a functional deficit. They study the effect of this increase of MSC on normal HSC's and observe no major impact of having increased numbers of PML null MSC's on HSC number. The authors then use the mice to explore a possible role of PML null MSC on leukemia initiating cells. Descriptive experiments in a CML model show a decrease in KLS cells in 2ndary recipients of unclear functional significance. Next they study in vitro co-culture of PML null or wt MSC with five different leukemia models. Leukemic cells are decreased in 3 of these models but not affected in two. Analysis is incomplete but the reduction in leukemic cell numbers appears to correlate with the reduction in KLS cells in the leukemia's. Studies of chemotherapy resistance in vivo using the MLL-AF9 model suggest a modest effect of PML null MSC on chemosensitivity in this model. An interesting change in c-Myc RNA in LSK cells from leukemia's incubated in PML null mice is shown but not commented on in the text. Figure 4 describes a very slight impairment of survival of leukemic cells after chemotherapy in Prrx1-Cre-PmlF/F recipients. In parallel experiments, the authors demonstrate well that decrease in PML expression leads to a decrease in expression of CXCL1 and IL6 from MSC and show that these cytokines may explain the impact of PML null MSC co-culture on leukemic cells. Overall, the manuscript is improved. The authors continue to muddle the difference between leukemic KSL cells and LIC. There are still two issues of clarity that should be addressed as below.

Major concerns:

- 1) I would re-iterate that although Figure 4 shows more data using MLL/AF9 model, no bona fide measurement (i.e. LDA assessment) of LIC is performed. This is despite the statement in the response to reviewers that such an assay was done. A limiting dilution analysis uses multiple cell doses in transplant to allow for calculation of a cell frequency (as I am sure the authors know). Using a small number of cells and calling it a limiting number of cells is a rather poor substitute. This is most significant as the investigator insists (for unclear reasons) on consistently referring to LIC.
- 2) For Figure 3b, I can not find a description of the number of cells (total leukemic or LSK) transplanted into recipient mice for the HoxA9/Meis model. Please provide. For clarity, this concern should be addressed before publication.
- 3) At what time after transplant was the engraftment data in Figure 4g generated? Are the differences seen here statistically significant? For clarity, this concern should be addressed before publication.

We thank both reviewers for their constructive and useful comments, which have been extremely helpful in improving our work. We have now addressed in the manuscript the additional concerns raised by the Reviewer #2.

Reviewer #1 (Remarks to the Author):

Authors properly addressed all my questions.

Reviewer #2 (Remarks to the Author):

This is a revised manuscript by Guarnerio and colleagues studying the hypothesis of a non-cell-autonomous role for PML expressed in MSC's in sustaining LICs in both CML and AML. I thank the authors for their revisions and for addressing my concerns. To again summarize, the authors have studied PML null and mice with Cre mediated deletion of PML in the MSC compartment. They demonstrate that loss of PML in MSC leads to an increase in MSC number without a functional deficit. They study the effect of this increase of MSC on normal HSC's and observe no major impact of having increased numbers of PML null MSC's on HSC number. The authors then use the mice to explore a possible role of PML null MSC on leukemia initiating cells. Descriptive experiments in a CML model show a decrease in KLS cells in 2ndary recipients of unclear functional significance. Next they study in vitro co-culture of PML null or wt MSC with five different leukemia models. Leukemic cells are decreased in 3 of these models but not affected in two. Analysis is incomplete but the reduction in leukemic cell numbers appears to correlate with the reduction in KLS cells in the leukemia's. Studies of chemotherapy resistance in vivo using the MLL-AF9 model suggest a modest effect of PML null MSC on chemosensitivity in this model. An interesting change in c-Myc RNA in LSK cells from leukemia's incubated in PML null mice is shown but not commented on in the text. Figure 4 describes a very slight impairment of survival of leukemic cells after chemotherapy in Prrx1-Cre-PmlF/F recipients. In parallel experiments, the authors demonstrate well that decrease in PML expression leads to a decrease in expression of CXCL1 and IL6 from MSC and show that these cytokines may explain the impact of PML null MSC co-culture on leukemic cells. Overall, the manuscript is improved. The authors continue to muddle the difference between leukemic KSL cells and LIC. There are still two issues of clarity that should be addressed as below.

Major concerns:

1) I would re-iterate that although Figure 4 shows more data using MLL/AF9 model, no bona fide measurement (i.e. LDA assessment) of LIC is performed. This is despite the statement in the response to reviewers that such an assay was done. A limiting dilution analysis uses multiple cell doses in transplant to allow for calculation of a cell frequency (as I am sure the authors know). Using a small number of cells and calling it a limiting number of cells is a rather poor substitute. This is most significant as the investigator insists (for unclear reasons) on consistently referring to LIC.

We understand the concern of the reviewer, and we agree that the term KLS cells, instead of LICs, might be more appropriate for the present manuscript. Accordingly, we have changed this through the manuscript. While, the conclusions of our experiments have not changed, the paper will now be more conservative and accurate in terms of taxonomy of the leukemic cells.

2) For Figure 3b, I can not find a description of the number of cells (total leukemic or LSK) transplanted into recipient mice for the HoxA9/Meis model. Please provide. For clarity, this concern should be addressed before publication.

We apologize for not having provided this relevant information. This information has been now included within the methods section of the revised manuscript.

3) At what time after transplant was the engraftment data in Figure 4g generated? Are the differences seen here statistically significant? For clarity, this concern should be addressed before publication.

We apologize for having missed relevant details about this experiment. We have now provided these details in the revised manuscript and figures.